

**1**  **The Influence of Carbon Cycling on Oxygen Depletion in North-Temperate Lakes**

**3**  Austin Delany[1], Robert Ladwig[1], Cal Buelo[1], Ellen Albright[1], Paul C. Hanson[1]

**4**  [1] Center for Limnology, University of Wisconsin-Madison, Madison, WI, USA

**5**  *Correspondence to*: Austin Delany (addelany@wisc.edu)

**7**  **Abstract.** Hypolimnetic oxygen depletion during summer stratification in lakes can lead to

**8**  hypoxic and anoxic conditions. Hypolimnetic anoxia is a water quality issue with many

**9**  consequences, including reduced habitat for cold-water fish species, reduced quality of

**10**  drinking water, and increased nutrient and organic carbon (OC) release from sediments. Both

**11**  allochthonous and autochthonous OC loads contribute to oxygen depletion by providing

**12**  substrate for microbial respiration; however, their relative importance in depleting oxygen

**13**  across diverse lake systems remains uncertain. Lake characteristics, such as trophic state,

**14**  hydrology, and morphometry are also influential in carbon cycling processes and may impact

**15**  oxygen depletion dynamics. To investigate the effects of carbon cycling on hypolimnetic

**16**  oxygen depletion, we used a two-layer process-based lake model to simulate daily

**17**  metabolism dynamics for six Wisconsin lakes over twenty years (1995-2014). Physical

**18**  processes and internal metabolic processes were included in the model and were used to

**19**  predict dissolved oxygen (DO), particulate OC (POC), and dissolved OC (DOC). In our

**20**  study of oligotrophic, mesotrophic, and eutrophic lakes, we found autochthony to be far more

**21**  important than allochthony to hypolimnetic oxygen depletion. Autochthonous POC

**22**  respiration in the water column contributed the most towards hypolimnetic oxygen depletion

**23**  in the eutrophic study lakes. POC water column respiration and sediment respiration had

**24**  similar contributions in the mesotrophic and oligotrophic study lakes. Differences in source



of respiration are discussed with consideration of lake productivity, hydrology, and
morphometry.



## 1 Introduction

Hypolimnetic oxygen depletion impacts lake ecosystems through its influences on lake
habitat and organic carbon (OC) cycling (Cole & Weihe 2016). In many lakes, oxygen
depletion results in hypoxia and even anoxia (Nürnberg 1995). Hypolimnetic anoxia reduces
habitat availability for cold-water fish species (Magee et al. 2019), reduces quality of
drinking water (Bryant et al. 2011), and can lead to elevated nutrient and OC release from
lake sediments (Hoffman et al. 2013, McClure et al. 2020). The formation of hypolimnetic
anoxia is associated with many internal and external lake characteristics, such as trophic
status (Rhodes et al., 2017; Rippey & McSorley, 2009), lake morphometry (Livingstone &
Imboden, 1996), and hydrology (Nürnberg 2004). An increase in the prevalence of
hypolimnetic anoxia and associated water quality degradation in temperate lakes indicates
the need to better understand how lake ecological processes interact with external forcing to
lead to the development of anoxia (Jane et al. 2021).

Hypolimnetic anoxia can occur when water column and sediment microbial respiration rates
exceed rates of oxygenation over an extended period. The conditions supporting oxygen
depletion are the outcomes of complex ecosystem processes and the interactions of the lake
with its climate and landscape settings (Jenny et al. 2016a, 2016b). Autochthonous OC
inputs vary considerably across trophic gradients and are a labile substrate for microbial
respiration that can contribute substantially to hypolimnetic anoxia (Müller et al. 2012,
2019). Allochthonous OC sources have also been shown to impact dissolved oxygen (DO)
and carbon dynamics in lakes by providing a more consistent and recalcitrant substrate for



respiration (Hanson et al. 2014, Solomon et al. 2015). Non-biological factors can be
important as well, such as the watershed loading of allochthonous OC, which can influence
the overall lability of OC in a lake and the rate of DO depletion (Hotchkiss et al. 2018).
Physical factors, such as stratification onset, water column stability, and vertical mixing, can
control the transport of DO from oxygen-rich upper layers to the lower layers of a lake, and
can therefore limit oxygen availability in the hypolimnion (Snortheim et al. 2017, Ladwig et
al. 2021). Lake morphometry can influence the spatial extents of stratified layers, which can
have profound effects on hypolimnetic volume and its capacity to hold DO as well as the rate
of sediment oxygen consumption, which can both influence anoxia onset in lakes
(Livingstone & Imboden 1996). Thus, the sources and lability of OC, lake morphometry, and
lake hydrodynamics all contribute to hypolimnetic oxygen depletion rates, making it an
emergent ecosystem property with a plethora of causal relationships to other ecologically
important variables.

Although previous studies have investigated contributions of allochthonous and
autochthonous OC to lake carbon cycling (Hanson et al. 2014, McCullough et al. 2018), the
effects on formation of hypolimnetic anoxia deserves further exploration (Hanson et al.
2015). The magnitude and relative balance of the sources of OC loads relates to hypolimnetic
anoxia across trophic and hydrology gradients (Rhodes et al., 2017; Rippey and McSorley,
2009, Hanson et al. 2014). These gradients affect the relative contributions of autochthony
and allochthony in a lake, which further control the lability and fates (respiration, burial,
export) of OC. The lability of OC relates to its form and its source (Hotchkiss et al. 2018,





Catalán et al. 2016). Autochthonous POC and DOC tend to be much more labile than
allochthonous OC (Amon & Brenner 1996, Thorpe & Delong 2002), thus understanding both
the forms of OC and their origins, in addition to their magnitudes, informs our understanding
of the controls over lake respiration. Quantifying the contribution of these different factors to
hypolimnetic anoxia is crucial to understanding its drivers across lakes and through time.

The availability of long-term observational data combined with process-based models
provides an opportunity to investigate OC sources and their control over the dynamics of lake
DO across multiple time scales. Long-term studies of lakes on regional and global scales
highlight how environmental trends can influence metabolic processes in lakes, and how
lakes can broaden our understanding of large-scale ecosystem processes (Richardson et al.
2017, Kraemer et al. 2017, Williamson et al. 2008). For example, long-term studies allow us
to investigate the impact that current and legacy conditions have on lake ecosystem function
in a given year (Carpenter et al. 2007). Process-based modeling has been used to investigate
metabolism dynamics and understand both lake carbon cycling (Hanson et al. 2004, Cardille
et al. 2007) and formation of anoxia (Ladwig et al. 2022); however, explicitly tying lake
carbon cycling and metabolism dynamics with long-term hypolimnetic DO depletion across a
variety of lakes remains largely unexplored. The combination of process-based modeling
with available long-term observational data, including exogenous driving data representative
of climate variability, can be especially powerful for recreating representations of long-term
lake metabolism dynamics (Staehr et al. 2010, Cardille et al. 2007).



In this study, our goal is to investigate OC source contributions to lake carbon cycling and
hypolimnetic oxygen depletion. We are particularly interested in the relative loads of
autochthonous and allochthonous OC to lakes and how they contribute to hypolimnetic DO
depletion across seasonal to decadal scales. We use a process-based lake metabolism model,
combined with daily external driving data and long-term limnological data, to study six lakes
within the North Temperate Lakes Long-Term Ecological Research network (NTL LTER)
over a twenty-year period (1995-2014). We address the following questions: (1) What are the
dominant sources of organic carbon that contribute to hypolimnetic oxygen depletion, and
how do their contributions differ across a group of diverse lakes over two decades? (2) How
do lake trophic state, hydrology, and morphometry influence the processing and fates of
organic carbon loads in ways that affect hypolimnetic dissolved oxygen?

**2 Methods**
**2.1 Study Site**
This study includes six Wisconsin lakes from the NTL-LTER program (Magnuson et al.
2006). Trout Lake (TR), Big Muskellunge Lake (BM), Sparkling Lake (SP), and Allequash
Lake (AL) are in the Northern Highlands Lake District of Wisconsin and have been regularly
sampled since 1981 (Magnuson et al. 2006). Lake Mendota (ME) and Lake Monona (MO)
are in southern Wisconsin and have been regularly sampled by the NTL-LTER since 1995
(NTL-LTER, Magnuson et al. 2006). The NTL-LTER provides a detailed description of each
lake (Magnuson et al. 2006). The six lakes span gradients in size, morphometry, landscape
setting, and hydrology, which creates diverse carbon cycling characteristics and processes



across these systems. TR and AL are drainage lakes with high allochthonous carbon inputs
from surface water, while BM and SP are groundwater seepage systems with allochthony
dominated by aerial OC inputs from the surrounding landscape (Hanson et al. 2014). All four
northern lakes (TR, AL, BM, SP) are surrounded by a forested landscape. ME and MO are
both eutrophic drainage lakes surrounded by an urban and agricultural landscape.
Morphometry, hydrology, and other information can be found in Table 1.
**Table 1.** Physical and biogeochemical characteristics of the study lakes. The table includes
lake area (Area), maximum depth (Zmax), hydrologic residence time (RT), mean annual
temperature (Temp), mean annual surface total phosphorus concentration (Mean TP), and
mean annual surface DOC (Mean DOC).

| Lake | Area (ha) | Zmax (m) | RT [3,4] (years) | Temp [2] (°C) | Mean TP [1] (µgL) | Mean DOC [1] (mgL) |
|---|---|---|---|---|---|---|
| Allequash Lake (AL) | 168.4 | 8 | 0.73 | 10.5 | 14 | 3.9 |
| Big Muskellunge (BM) | 396.3 | 21.3 | 5.1 | 10.5 | 7 | 3.8 |
| Sparkling Lake (SP) | 64 | 20 | 8.88 | 10.6 | 5 | 3.12 |
| Trout Lake (TR) | 1607.9 | 35.7 | 5.28 | 9.8 | 5 | 2.8 |
| | | | | | | |
| Mendota (ME) | 3961 | 25.3 | 4.3 | 12.5 | 50 | 5.6 |
| Monona (MO) | 1324 | 22.5 | 0.7 | 13.8 | 47 | 5.8 |

1 - Magnuson et al. (2020, 2006)
2 - Magnuson et al. (2022)



3 - Hunt et al. (2013)
4 - Webster et al. (1996)



## 2.2 Driving Data and Limnological Data

Most driving data for the model is provided by the "Process-based predictions of water

temperature in the Midwest US" USGS data product (Read et al. 2021). This includes lake

characteristic information such as lake area and hypsometry, daily modeled temperature

profiles, ice flags, meteorology data, and solar radiation for the six study lakes. Derived

hydrology data is used in calculating daily OC loading and outflow for the study lakes.

Hydrology for the northern lakes is taken from Hunt & Walker (2017), which was estimated

using a surface and groundwater hydrodynamic model. Hydrology for ME is taken from

Hanson et al. (2020), which used the Penn State Integrated Hydrologic Model (Qu & Duffy

2007). We found that the derived discharge data for ME, TR, AL, and SP was approximately

20-50% higher than previously reported values (Hunt et al. 2013, Webster et al. 1996),

depending on the lake, while hydrology in BM was approximately 25% too low (Hunt et al.

2013). To accommodate this issue, we adjusted total annual hydrological inputs to match

published water residence times for each lake (Table 1), while retaining temporal

hydrological patterns. NTL-LTER observational data are interpolated to estimate daily

nutrient concentration values, which are used in calculating daily primary production in the

model (Magnuson et al. 2020).






The NTL-LTER observational data used to calibrate and validate the model for the six lakes
include DO, DOC, and Secchi depth (Magnuson et al. 2020, Magnuson et al. 2022).
Saturation values for DO and gas exchange velocity used in calculating atmospheric
exchange for DO are calculated using the "o2.at.sat.base" and using the Cole and Caraco gas
exchange method from the "K600.2.KGAS.base" function within the USGS
"LakeMetabolizer" package in R (Winslow et al. 2016).

**2.3 The Model**
The goal of our model is to use important physical and metabolic processes involved in the
lake ecosystem carbon cycle to best predict DO, DOC, and POC, while keeping the model
design simple in comparison with more comprehensive water quality models (e.g., Hipsey et
al. 2019). We ran our model with a daily time step over a twenty-year period for each lake
and includes seasonal physical dynamics, such as lake mixing, stratification, and ice cover
from Read et al. 2021. Throughout each year, the model tracks state variables and fluxes in
the lake for each day (Fig. 1). These state variables include DO and the labile and recalcitrant
components of particulate organic carbon (POC) and dissolved organic carbon (DOC).
During stratified periods, the state variables and fluxes for the epilimnion and hypolimnion
are tracked independently. Atmosphere, sediments, and hydrologic inputs and outputs are
boundary conditions.

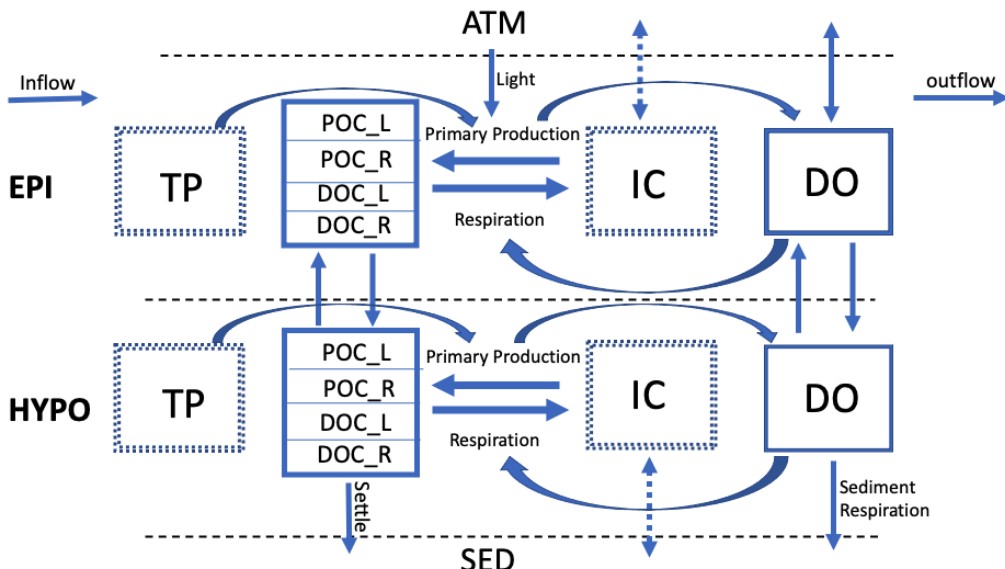

**Figure 1.** Conceptual lake model showing state variables (boxes) and fluxes (arrows).  The
model has two thermal layers under stratified conditions, as shown here, and tracks state
variables separately for each layer. The sediment (SED), atmosphere (ATM), inflow and
outflow are system boundaries. The state variables included are DO (dissolved oxygen),
DOC_L (labile dissolved organic carbon), DOC_R (recalcitrant dissolved organic carbon),
POC_L (labile particulate organic carbon), and POC_R (recalcitrant particulate organic
carbon). Inorganic carbon (IC) is not tracked in the model and is assumed to be a non-
limiting substrate to primary production. Observed total phosphorus (TP) is used as a driving
variable for primary production in the model.

**2.3.1 Stratification Dynamics**
Lake physical dynamics are taken from the output of a previous hydrodynamic modeling
study on these same lakes over a similar time period (Read et al. 2021), which used the
General Lake Model (Hipsey et al. 2019). Before running the metabolism model, a
thermocline depth for each time step is estimated using derived temperature profiles for each
lake (Read et al. 2021) by determining the center of buoyancy depth (Read et al. 2011). After
calculating the thermocline depth, the volumes and average temperatures for each layer, and
the specific area at thermocline depth are determined using lake-specific hypsography. The



criteria for stratification include a vertical density gradient between the surface and bottom
layer of at least 0.05 kg m$^{-3}$, an average water column temperature above 4 °C, and the
presence of a derived thermocline (Ladwig et al. 2022). For any day that does not meet all of
these criteria, the water column is considered to be fully mixed. The thermocline depth
values are smoothed using a moving average with a window size of 14 days to prevent large
entrainment fluxes that can destabilize the model at very short time scales when thermal
strata are shallow. During mixed periods, the entire lake is treated as the epilimnion, and a
separate hypolimnion is not incorporated into the model dynamics. Ice cover in the model is
determined using the "ice flag" provided in the derived temperature profile data from Read et
al. (2021). Our metabolism model does simulate under-ice conditions, however we do not
include the presence of inverse stratification during winter periods.

**2.3.2 External Lake and Environment Physical Fluxes**
Atmospheric exchange of DO, external loading of OC, and outflow of OC are the three
environmental boundary fluxes accounted for in the water quality model (Table 3 Eq. 9-11).
The gas exchange velocity for atmospheric exchange is determined using the Cole and
Caraco model (1998) and is calculated using the LakeMetabolizer R package (Winslow et al.
2016). Oxygen saturation values are also calculated using this package. During ice covered
conditions, we assume that the atmospheric exchange value is ten percent of the value during
non-ice covered conditions based on sea ice gas exchange estimates (Loose and Schlosser,

253     2011).




For the northern lakes (TR, AL, BM, SP), we use the allochthonous OC load and recalcitrant
OC export values from Hanson et al. (2014) to calibrate total annual allochthonous OC load
and recalcitrant OC export in our model. We specifically use the allochthonous OC load
values in this study to assist in the manual calibration of inflow recalcitrant POC and DOC
concentrations for each lake. For the southern lakes (ME, MO), we also use derived
hydrology information (Hanson et al. 2020), but only for discharge that is the inflow for ME.
We assume for ME and MO that evaporation from the lake surface is approximately equal to
precipitation on the lake surface and that groundwater inputs and outputs to the lake are a
small part of the hydrologic budgets (Lathrop & Carpenter 2014). Therefore, ME outflow is
assumed to be equal to ME inflow. ME is the predominant hydrologic source for MO
(Lathrop & Carpenter 2014), thus, MO inflow is assumed to be equal to ME outflow, and
MO outflow is assumed to be equal to MO inflow. ME allochthonous load is calibrated based
on model fitting and observational data (Hart et al. 2019). MO inflow concentrations are
equivalent to the in-lake epilimnetic concentrations of OC from ME at each model time step.
The OC loads for MO are calibrated based on the total allochthonous load found in
McCullough et al. 2018.
**Table 2.** Equations for the model, organized by state variables, [*DO* (dissolved oxygen),
*DOC_L* (labile dissolved organic carbon), *DOC_R* (recalcitrant dissolved organic carbon),
*POC_L* (labile particulate organic carbon),  *POC_R* (recalcitrant particulate organic carbon),
*Secchi*] and relevant fluxes. *Note:* The entrainment flux (*Entr*) is only included during
thermally stratified periods. The inflow (*IN*) and outflow (*OUT*) fluxes are not included in
the calculations for the hypolimnetic layer. Atmospheric gas exchange of dissolved oxygen
(*AtmExch*) is not included for the hypolimnetic DO calculation. Normalized total phosphorus
is represented by (*TP_norm*). The volume (*V*) term represents the respective lake layer volume,



or the discharge volume for the inflow and outflow equations. Terms not defined here are
included in Table 3.

| ***State Variables*** | |
|---|---|
| DO [gDO] $$\frac{dDO}{dt} = NPP * O2_{convert}) + AtmExch + Entr_{DO} - (R_{sed} * O2_{convert}) - (R_{wc} * O2_{convert})$$ | (1) |
| DOC_L [gC] $$\frac{dDOC_L}{dt} = (NPP * (1 - C_{NPP})) + IN_{DOCL} + Entr_{DOCL} - R_{DOCL} - OUT_{DOCL}$$ | (2) |
| DOC_R [gC] $$\frac{dDOC_R}{dt} = IN_{DOCR} + Entr_{DOCR} - OUT_{DOCR} - R_{DOCR\ Epi}$$ | (3) |
| POC_L [gC] **Mixed and Epi:** $\frac{dPOC_L}{dt} = (NPP_{Epi} * C_{NPP}) + IN_{POCL} + Entr_{POCL} - R_{POCL\ Epi} - Settle_{POCL\ Epi} - OUT_{POCL}$ | (4) |
| **Hypo:** $\frac{dPOC_L}{dt} = (NPP_{Hypo} * C_{NPP}) + Settle_{POCL\ Epi} - Settle_{POCL\ Hypo} - R_{POCL\ Hypo} - Ent_{POCL}$ | (5) |
| POC_R [gC] **Mixed and Epi:** $\frac{dPOC_R}{dt} = IN_{POCR} + Entr_{POCR} - OUT_{POCR} - R_{POCR\ Epi} - Settle_{POCR\ Epi}$ | (6) |
| **Hypo:** $\frac{dPOC_R}{dt} = Settle_{POCR\ Epi} - Settle_{POCR\ Hypo} - R_{POCR\ Hypo} - Entr_{POCR}$ | (7) |
| Secchi [m] $$Secchi = \frac{1.7}{K_{LEC}}$$ | (8) |
| ***Fluxes*** | |
| Atm exchange [gDO d⁻¹] $$AtmExch = K_{DO} * (DO_{sat} - DO_{prediction}) * Area_{sfc}$$ | (9) |
| Inflow [gC d⁻¹] $$IN = Carbon\ Concentration_{inflow} * V_{inflow}$$ | (10) |
| Outflow [gC d⁻¹] $$OUT = Carbon\ Concentration_{outflow} * V_{outflow}$$ | (11) |
| Net Primary Productivity [gC d⁻¹] $$NPP = Pmax * (1 - e^{(-IP*\frac{Light}{Pmax})}) * TP_{norm} * \theta_{NPP}^{(T-20)} * V$$ | (12) |
| Respiration [gC d⁻¹] $$R = Carbon\ Pool * r_{rate} * \theta_{Resp}^{(T-20)} * \frac{DO_{Concentration}}{DO_{1/2} + DO_{Concentration}}$$ | (13) |
| Sediment Respiration [gC d⁻¹] $$R_{sed} = r_{sed} * \theta_{Resp}^{(T-20)} * \frac{DO_{Concentration}}{DO_{1/2} + DO_{Concentration}} * Area_{sed}$$ | (14) |
| POC settle [gC d⁻¹] $$Settle = (POC\ Pool * K_{POC}) * \frac{Area}{V}$$ | (15) |
| Entrainment [gC d⁻¹] $$V_{Entr} = V_{epi}(t) - V_{epi}(t-1)$$ $V_{Entr} > 0$ (Epilimnion growing) $$Entr = \frac{V_{Entr}}{V_{Hypo}} * Carbon\ Pool_{Hypo}$$ $V_{Entr} < 0$ (Epilimnion shrinking) $$Entr = \frac{V_{Entr}}{V_{Epi}} * Carbon\ Pool_{Epi}$$ | (16) (17) (18) |



| Light [W m$^{-2}$]$$Light = \int_{z_1}^{z_2}(I_{z_1} * e^{-(K_{LEC} * z)}) \, dz * (1 - \alpha)$$ | (19) |
|---|---|
| Light Extinction Coefficient [Unitless]$$K_{LEC} = LEC_{water} + (LEC_{POC} * ((\tfrac{POCL}{V}) + (\tfrac{POCR}{V}))) + (LEC_{DOC} * ((\tfrac{DOCL}{V}) + (\tfrac{DOCR}{V})))$$ | (20) |



### 2.3.3 Internal Lake Physical Fluxes

The two in-lake physical fluxes included in the model are POC settling and entrainment of all

state variables. POC settling is the product of a sinking rate (m d$^{-1}$) and the respective POC

pool (g), divided by the layer depth (m) (Table 3 Eq. 15). Sinking rates are either borrowed

from literature values (Table 3) or fit during model calibration (see below). Entrainment is

calculated as a proportion of epilimnetic volume change (Table 2 Eq. 17-18). A decrease in

epilimnetic volume shifts mass of state variables from the epilimnion into the hypolimnion,

and an increase in volume shifts mass from the hypolimnion to the epilimnion.

**Table 3.** Model Parameters, grouped by static and free parameters

| Parameter | Abbreviation | Value | Units | Comments |
|---|---|---|---|---|
| Respiration rate of DOCR | $r_{DOCR}$ | 0.001 | $day^{-1}$ | (Hanson et al., 2011) |
| Respiration rate of POCR | $r_{POCR}$ | 0.005 | $day^{-1}$ | Based on ranges provided in (Hanson et al. 2004) and estimated from manual model fitting |
| Respiration rate of POCL | $r_{POCL}$ | 0.2 | $day^{-1}$ | Based on ranges provided in (Hipsey et al. 2019) and estimated from manual model fitting |
| Michaelis-Menten DO half saturation coefficient | $DO_{1/2}$ | 0.5 | $g \ m^{-3}$ | Determined through manual model fitting |
| Conversion of Carbon to Oxygen | $O2_{convert}$ | 2.67 | Unitless | Mass Ratio of C:O |
| Fitting coefficient for Light extinction of water | $LEC_{water}$ | 0.125 | $m^{-1}$ | Manually calibrated based on observed Secchi Depth ranges for the study lakes |
| Fitting coefficient for Light extinction of DOC | $LEC_{DOC}$ | 0.02 - 0.06 | $m^2 g^{-1}$ | Manually calibrated based on observed Secchi Depth ranges for the study lakes |



| | | | | |
|---|---|---|---|---|
| Fitting coefficient for Light extinction of POC | $LEC_{POC}$ | 0.7 | $m^2 g^{-1}$ | Manually calibrated based on observed Secchi Depth ranges for the study lakes |
| Ratio of DOC to POC production from NPP | $C_{NPP}$ | 0.8 | Unitless | Hipsey et al. 2019 Hipsey et al. 2019 |
| Atmospheric gas exchange adjustment during ice covered conditions | $C_{winter}$ | 0.1 | Unitless | Estimated from manual model fitting and ranges provided in (Loose & Schlosser, 2011) |
| Coefficient of light transmitted through ice | $C_{ice}$ | 0.05 | Unitless | Based on ranges provided in (Lei et al. 2011) and estimated from manual model fitting |
| Settling velocity rate of POC_R | $K_{POCR}$ | 1.2 | $m\ day^{-1}$ | Based on ranges found in (Reynolds et al.1987) and estimated from manual model fitting |
| Settling velocity rate of POC_L | $K_{POCL}$ | 1 | $m\ day^{-1}$ | Based on ranges ranges found in (Reynolds et al.1987) and estimated from manual model fitting |
| Temperature scaling coefficient for NPP | $\theta_{NPP}$ | 1.12 | Unitless | Based on Q10 of 2 principles and estimated from manual model fitting |
| Temperature scaling coefficient for Respiration | $\theta_{Resp}$ | 1.04 | Unitless | Based on Q10 of 2 principle and estimated from manual model fitting |
| Albedo | $\alpha$ | 0.3 | Unitless | Global average (Marshall & Plumb, 2008) |
| Maximum Daily Productivity | $Pmax$ | 0.5-5 | $g\ m^{-3}day^{-1}$ | Range based on mean productivity values from Wetzel (2001) and manual model fitting |
| Recalcitrant DOC inflow concentration | $DOCR_{inflow}$ | 5-10 | $g\ m^{-3}$ | Based on ranges found in (Hanson et al. 2014, McCullough et al. 2018, Hart et al. 2017) and manual model fitting |
| Recalcitrant POC inflow concentration | $POCR_{inflow}$ | 2-5 | $g\ m^{-3}$ | Based on ranges found in (Hanson et al. 2014, McCullough et al. 2018, Hart et al. 2017) and manual model fitting |
| **Free parameters** | | | | |
| Slope of the irradiance/productivity curve | $IP$ | 0.055, 0.020 | $gCd^{-1}(Wm^{-2})^{-1}$ | Based on ranges found in (Platt et al. 1980) and fit for each lake region independently (South, North) |
| Respiration rate of sediments | $r_{sed}$ | $0.1 - 0.4$ | $day^{-1}$ | Fit independently for each lake |
| Respiration rate of DOCL | $r_{DOCL}$ | 0.015, 0.020 | $day^{-1}$ | Fit for each lake region independently (South, North) |






### 2.3.4 Internal Lake Metabolism Fluxes

The metabolism fluxes in the model are net primary production (NPP) and respiration (R).
Respiration includes water column respiration for each OC state variable in the epilimnion
and hypolimnion and is calculated at each time step as the product of the OC state variable
and its associated first order decay rate (Table 2, Eq. 13). Sediment respiration for the
hypolimnion during stratified periods and the epilimnion (entire lake) during mixed periods
is a constant daily rate that is individually fit for each lake. We assume inorganic carbon is
not a limiting carbon source. In the model, we consider any DO concentration less than 1 g
DO $m^{-3}$ to be anoxic (Nürnberg 1995).

NPP is tracked in both the epilimnion and hypolimnion. NPP is a function of light, total
phosphorus concentration, temperature, a maximum productivity coefficient, and a slope
parameter defining the irradiance and productivity curve (Table 2 Eq. 12). Average light in a
layer is calculated for each day and is dependent on the depth of a layer and the light
extinction coefficient (Table 2 Eq. 19). During ice covered conditions, average light is
assumed to be five percent of the average non-ice covered value (Lei et al. 2011). Total
phosphorus concentration in a layer is from observational data for each lake interpolated to
the daily time scale. The interpolated values are then normalized for each individual lake to
drive NPP. These values are normalized so that differences among lakes are only present in
the IP and $P_{max}$ parameters. The Arrhenius equation provides temperature control for NPP,
and we determined through model fitting a $\theta$ of 1.12. OC derived from NPP is split between



particulate and dissolved labile OC production, with eighty percent produced as POC and
twenty percent produced as DOC. This ratio was determined through model fitting and is
similar to previously reported values (Hipsey et al. 2019).

Epilimnetic and hypolimnetic water column respiration is tracked independently for each OC
pool in the model. During mixed periods, there are four OC pools – DOCR, DOCL, POCR,
POCL. During stratified periods, those pools are split into a total of eight pools that are
tracked independently for the epilimnion and hypolimnion. Respiration is calculated as a
product of the mass of a respective variable, a first order decay rate coefficient, temperature,
and oxygen availability (Table 2 Eq. 13). The respiration decay rate coefficients are based on
literature values (Table 3) or were fit during model calibration. An Arrhenius equation is
used for temperature control of respiration, with $\theta$ equal to 1.04, which was determined
through manual model fitting. The respiration rates are also scaled by oxygen availability
using the Michaelis-Menten equation with a half saturation coefficient of 0.5 g DO m$^{-3}$, such
that at very low DO concentrations, the respiration flux approaches zero.

Sediment respiration is calculated from a constant daily respiration rate coefficient, adjusted
for temperature and oxygen availability, using the Arrhenius and Michaelis-Menten
equations, respectively (Table 2 Eq. 14). The mass of sediment OC is not tracked in the
model. During stratified periods, we assume that the majority of epilimnetic sediment area is
in the photic zone, and therefore has associated productivity from macrophytes and other
biomass. It is assumed that this background productivity and sediment respiration are of





similar magnitude and inseparable from water column metabolism, given the observational
data. Therefore, epilimnetic sediment respiration is not accounted for in the model during
stratified conditions. During mixed conditions, we assume that sediment respiration is active
on all lake sediment surfaces, which are assumed to be equivalent in area to the total surface
lake area. During stratified periods, we use the area at the thermocline as the sediment area
for calculating hypolimnetic sediment respiration.

**2.3.5 Other in-lake calculations and assumptions**
We calculate a total light extinction coefficient (LEC) for the epilimnion and hypolimnion.
The total LEC for each layer is calculated by multiplying the dissolved and particulate
specific LEC values with their respective OC state variable concentrations, combined with a
general LEC value for water (Table 2 Eq. 20). This total LEC value is used to calculate a
daily estimate of Secchi depth (Table 2 Eq. 8). The coefficients for the light extinction of
water, DOC, and POC are manually calibrated based on observed Secchi depth ranges for the
study lakes (Table 3, SI Table 5).

**2.4 Model calibration and validation**
The model was run for twenty years from 1 January 1995 to 31 December 2014. This period
was chosen due to an absence of hydrologic data for the northern lakes after 2014 and
because consistent observational data weren't available for the southern lakes until 1995. The
first 15 years of the model output was used for calibration and the last 5 years were used for
model validation. We chose the first 15 years for calibration because the observational data



were relatively stable and were not indicative of any large trends in ecosystem processes, as
opposed to the last five years which showed slightly more model deviation from DOC
observational data in the southern lakes (SI Fig. 2).

Initial conditions for each lake state variable are based on literature values or lake
observational data (SI Table 5). The model is initialized on January 1st of the first year, so
we set the initial labile POC mass to zero under the assumption that the concentration is low
in the middle of winter. The initial DO value is set to the saturation value based on the
conditions of the initial model run day and is calculated using the LakeMetabolizer R
package (Winslow et al. 2016).

**2.5 Model Fitting and Parameter Uncertainty Estimation**
The free parameters in the model are the slope of the irradiance/productivity curve (IP), the
respiration rate of labile DOC (Resp_DOCL), and the respiration rate of the hypolimnetic
sediments (Resp_sed) (Table 3). These were in part chosen due to the high uncertainty
around the parameter values, and our assumptions that they have a higher impact on
ecosystem dynamics in the model. Optimized values and uncertainties for each free
parameter and lake are included in SI Table 4.

IP controls the amount of productivity in low light scenarios, and fitting the parameter helps
to calibrate productivity during ice-covered winter conditions as well as during times of high
OC concentrations in the epilimnion. Resp_DOCL controls the seasonal dynamics of DOC in

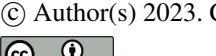



a lake and treating it as a free parameter helps capture the across-lake variability in DOC
processes related to variations in landscape, hydrology, and productivity. Resp_sed is
important for controlling hypolimnetic oxygen depletion in lakes and is related to lake
productivity and associated legacy OC in each lake. The Resp_sed parameter also helps
adjust burial rates for the study lakes. We fit unique IP and Resp_DOCL values for the
southern lake region (ME, MO), and unique values for the northern region (TR, AL, BM,
SP). Resp_sed is individually fit for each lake.

We manually optimized free parameters by manually adjusting them over their respective
ranges to find the parameter values that returned the smallest model residuals (SI Table 4).
Automated optimization proved too computationally demanding. To gain a better sense of
the contributions of parameter uncertainty in the model, we created parameter uncertainty
distributions using standard deviations of 20% of the estimated parameter value. To evaluate
the influence of parameter uncertainties on model predictions, the parameter distributions are
randomly sampled over 100 model iterations to create the uncertainty bounds for all
predictions of model state variables and fluxes.

During the model fitting, errors in modeled DO, DOC, and Secchi depth are weighted
equally in the southern lakes. In the northern lakes, fitting Secchi depth was challenging.
Initial model fits revealed that patterns in observed Secchi did not show regular seasonality
and were highly stochastic. Therefore, we use a moving average on observational data and
predictions of Secchi depth and calculate the residuals as the difference between the two



averaged time series. This is done to remove stochasticity from the observational data and fit
the model predictions to the average observed Secchi value. We use a moving average
window of 15 observations because we want to capture the average annual Secchi depth
trend, and there are roughly 15 observations per year.

**3 Results**
**3.1 Model Fit to Ecosystem States**
Model predictions of DO reproduce observed seasonal variability well. RMSE values
presented here represent model error combined over both the validation and calibration
periods (see Supplementary Material: Table S1 for calibration and validation specific RMSE
values). State variables are presented with truncated time ranges for visual clarity (see
Supplementary Material: Fig. S1-S3 for full time series). Epilimnetic DO generally has lower
RMSE than DO in the hypolimnion (Fig. 2). In the epilimnion, RMSE ranges from 0.73 g
DO m$^{-3}$ (TR) to 2.11 g DO m$^{-3}$ (ME), and in the hypolimnion, RMSE ranges from 1.20 g DO
m$^{-3}$ (TR) to 2.69 g DO m$^{-3}$ (AL). In the southern lakes, modeled values reach anoxic levels
and generally follow the DO patterns recorded in the observed data (Fig. 2a-b).
Observational data for the northern lakes show an occasional late summer onset of anoxia,
and these events are generally captured in the model output. A late summer spike in
hypolimnetic DO predictions commonly occurs as well, which is likely a model artifact
caused by the reduction of hypolimnetic volumes to very small values over short time periods
prior to fall mixing. Reduction to small volumes, coincident with modest fluxes due to high
concentration gradients, result in transient high concentrations. Overall, the goodness-of-fit





of hypolimnetic DO in our study lakes does not seem to follow any regional or lake
characteristic patterns.

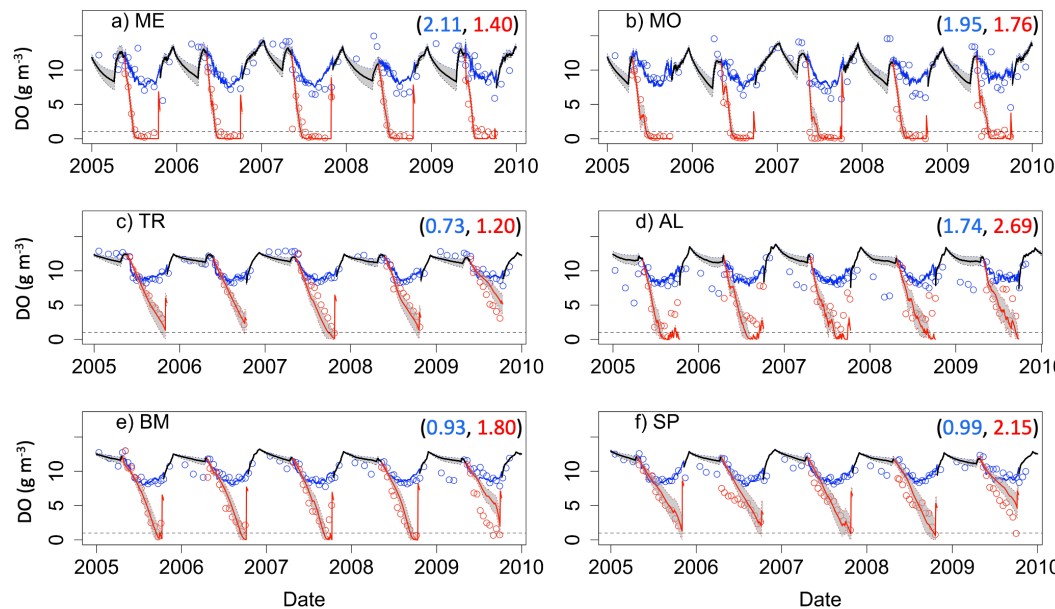

**Figure 2.** Dissolved oxygen (DO) time series for the years, 2005-2010, for the six study
lakes (a-f). Model predictions are represented by lines, and circles represent the observational
data. Epilimnetic DO values are blue and Hypolimnetic DO values are red. Fully mixed
periods for the lake are indicated by a single black line. RMSE values (epilimnion,
hypolimnion; g m$^{-3}$) are included in the upper right of each panel. Uncertainty is represented
by gray shading.


The two southern lakes (ME, MO) have DOC RMSE values equal to or greater than 1.00 g C
m$^{-3}$, while the RMSE for northern lakes ranges from 0.28 g C m$^{-3}$ (TR) to 0.60 g C m$^{-3}$ (AL)
(Fig. 3). Observational data in both southern lakes indicate a decrease in DOC concentration
beginning around 2010, which is largely missed in the model predictions (Fig.3a-b,
Supplementary Material: Fig. S2a-b) and cause an overestimation of DOC by about 1-2 g C
m$^{-3}$. However, model predictions converge with observed DOC toward the end of the study
period (Supplementary Material: Fig. S2a-b). In AL, the seasonal patterns of modeled DOC
are smaller in amplitude than the observational data (Supplementary Material: Fig. S2d).

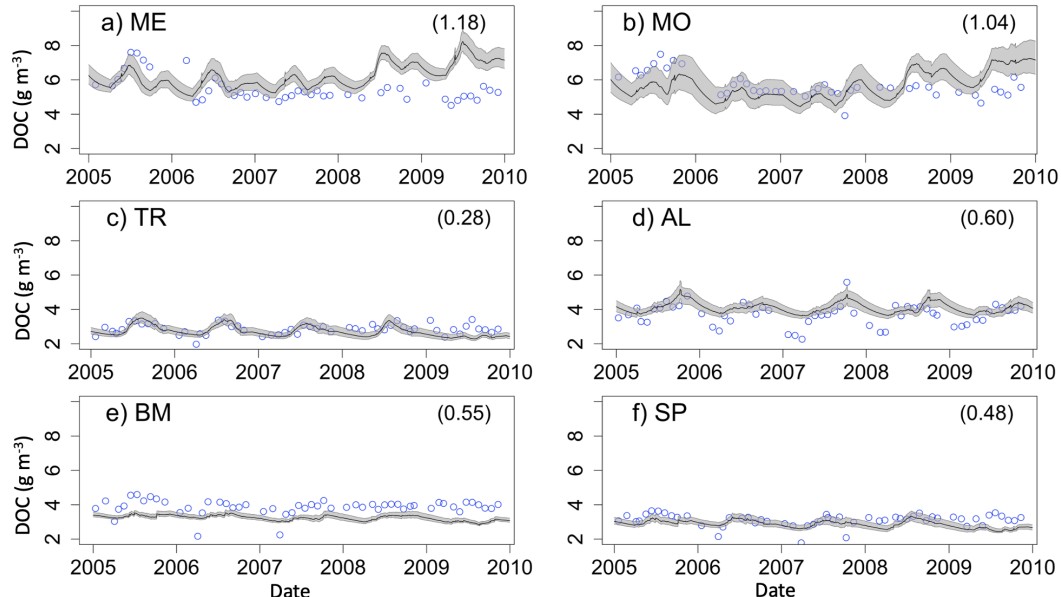

**Figure 3.** Epilimnetic dissolved organic carbon (DOC) time series for the years, 2005-2010,
for the six study lakes (a-f). Model predictions are represented by lines, and circles represent
the observational data. RMSE values are included for each lake (g C m$^{-3}$). Uncertainty is
represented by gray shading.

Secchi depth predictions reproduce the mean and seasonal patterns in most lakes (Fig. 4).
Although the model produced annual cycles of Secchi depth that generally covered the range
of observed values, short term deviations from annual patterns in the observed data are not
reproduced. The timing of minima and maxima Secchi depth sometimes differed between
predicted and observed values for the northern lakes. In addition, winter extremes in
observed Secchi depth are not always reproduced by the model, which is especially evident
for ME (Fig. 4a). However, winter observational data for Secchi are more sparse than other
seasons.

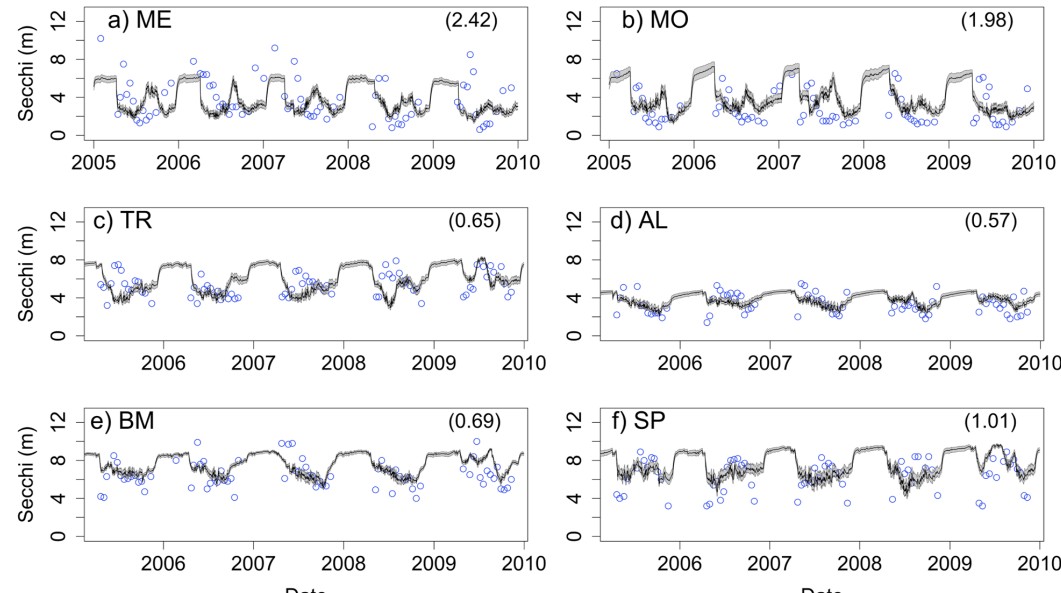

**Figure 4.** Secchi depth time series for the years, 2005-2010, for the six study lakes (a-f).
Model predictions are represented by lines, and circles represent the observational data.
RMSE values are included for each lake (m). Uncertainty is represented by gray shading.

**3.2 Ecosystem Processes**
The mean annual OC budgets of all six lakes show large differences in the sources and fates
of OC among lakes (Fig. 5; Supplementary Material: Table S3). Autochthony is the dominant
source of OC for all study lakes. Water column respiration is the largest portion of whole-
lake respiration in ME, MO, TR, SP, and BM. Sediment respiration contributions are a lower
proportion of total respiration in ME, MO, and TR (mean of 15.0%), and are slightly higher
in BM and SP (mean of 23.7%). AL has a more even distribution of OC fates. OC burial
amounts also vary across the study lakes, with the highest percentage in AL (26.4%), and
lowest in SP (2.5%).




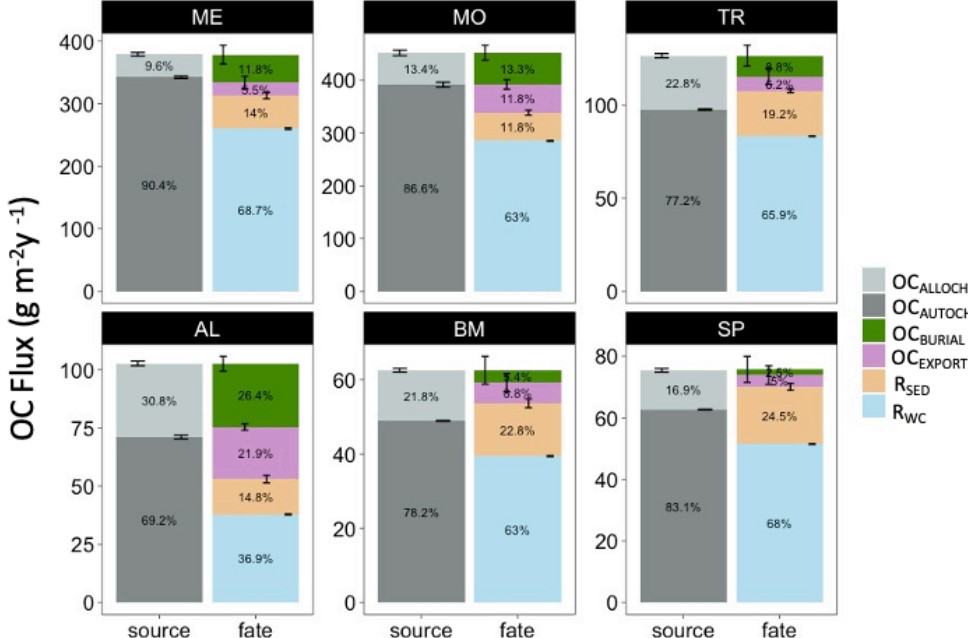

**Figure 5.** Total annual budget, sources (left stacked bars) and fates (right stacked bars), of
organic carbon (OC) in each lake over the study period. The OC sources include
allochthonous OC (OC$_{ALLOCH}$) and autochthonous OC (OC$_{AUTOCH}$). The OC fates include
burial of OC (OC$_{BURIAL}$), export of OC (OC$_{EXPORT}$), sediment respiration of OC (R$_{SED}$), and
water column respiration of OC (R$_{WC}$). Standard error bars for the annual means are
indicated for each source and fate as well. Note that the magnitudes of the y-axis differ
among the lakes.

The study lakes show inter-annual variation in trophic state, as quantified by NEP (Fig. 6).
Total respiration (water column and sediment) exceeds autochthony in SP, BM, and TR,
indicating net heterotrophy for these systems. The remaining lakes (ME, MO, AL) are net
autotrophic. The southern lakes (ME, MO) were net autotrophic (positive NEP) for the
majority of the study years but became less autotrophic over the last five years of the study
period (2010-2014). TR, BM, and SP were mostly net heterotrophic (negative NEP) over the





study period with a few brief instances of net autotrophy. The strongest autotrophic signal for
these lakes occurred around 2010. AL is mostly net autotrophic over the study period but had
lower average NEP than the southern lakes. AL also experienced a negative NEP in 2012.
ME, MO, and AL all have negative trends in NPP, but only ME was significant (p_value <
0.1, Mann-Kendall test) (SI Table 2). Of these three lakes, ME and AL also have decreasing
significant trends in annual total phosphorus concentration (SI Table 2). No significant trends
were found for NPP or total phosphorus in the other lakes (MO, TR, BM, SP).

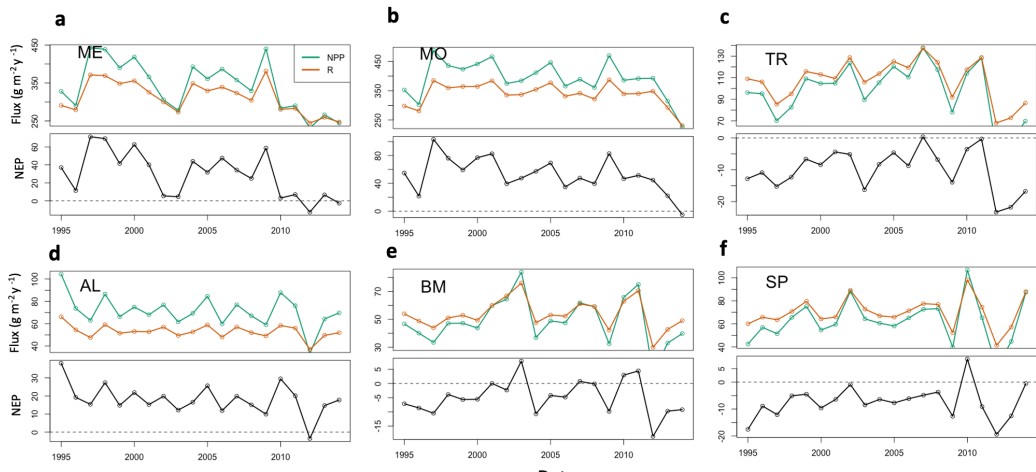

**Figure 6.** Time series of lake Net Primary Production (green) and Total Respiration (red)
(top panels), and Net Ecosystem Production (NEP, bottom panels) for the six lakes: (a) Lake
Mendota; (b) Lake Monona; (c) Trout Lake; (d) Allequash Lake; (e) Big Muskellunge Lake,
and; (f) Sparkling Lake. Fluxes are in units of $gC\ m^{-2}y^{-1}$
Hypolimnetic DO consumption during stratified periods was due to the two components of
hypolimnetic respiration, hypolimnetic water column respiration and hypolimnetic sediment
respiration. Water column respiration contributes more than sediment respiration to total
hypolimnetic respiration in the southern lakes compared to the northern lakes, with the
exception of TR, where cumulative water column respiration is much larger than cumulative



sediment respiration. In ME and MO, the mass of summer autochthonous POC entering the
hypolimnion is similar to the total hypolimnetic OC mass respired for the beginning of the
stratified period (Fig. 7a-b; green line). Later in the stratified period, an increase in
epilimnetic POC and associated settling exceeds total hypolimnetic respiration (Fig. 7a-b;
green hashed area). This is due, in part, to lower respiration rates that occur once DO (gray
line) has been fully depleted, which occurs in early July for ME and late June for MO. In TR
and SP the total hypolimnetic respiration slightly exceeds autochthonous POC inputs over the
duration of the stratified period, indicating the importance of allochthony in these systems
(Fig. 7c,f). BM shows that autochthonous POC entering the hypolimnion and total
hypolimnetic respiration are similar throughout the stratified period (Fig. 7d). AL is the only
lake to have autochthonous POC inputs consistently larger than total hypolimnetic respiration
during the stratified season. All lakes show that summer allochthonous POC entering the
hypolimnion is a small contribution to the overall hypolimnetic POC load.

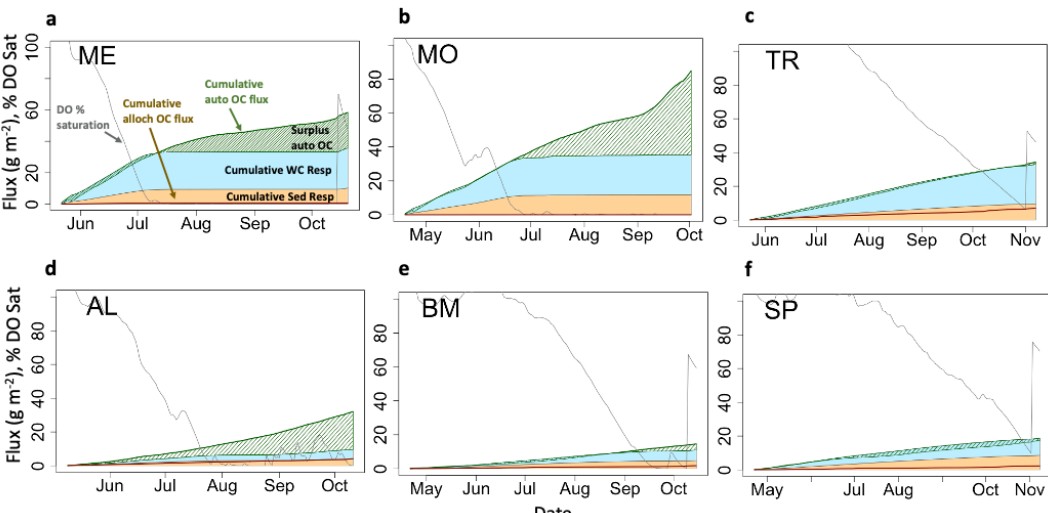

**Figure 7.** Hypolimnetic dissolved oxygen, allochthonous (alloch) and autochthonous (auto)
organic carbon loading, and respiration dynamics during one stratified period (2005) for each





lake. Fluxes are cumulative $gC\ m^{-2}$ and DO is presented as percent saturation. Labels are in
panel (a). Note that the cumulative water column (WC) and sediment (Sed) respiration fluxes
are stacked, while other cumulative fluxes are not.

Respiration of autochthonous POC and sediment respiration account for most of the total
hypolimnetic respiration in all lakes (Fig. 8). Respiration of DOC accounts for a relatively
small proportion of total respiration. Total hypolimnetic respiration is higher in the southern
lakes than the northern lakes. TR has the highest amount of hypolimnetic respiration for the
northern lakes, and AL and BM have the least amounts of hypolimnetic respiration. Water
column respiration contributed the most towards total hypolimnetic respiration in ME, MO,
and TR. Sediment respiration and water column respiration contributed similar proportions
towards total hypolimnetic respiration in BM, SP, and AL. As total respiration across lakes
increases, a larger proportion of that respiration is attributable to respiration of POC in the
water column. DOC water column respiration was the smallest proportion of total
hypolimnetic respiration in all six study lakes.




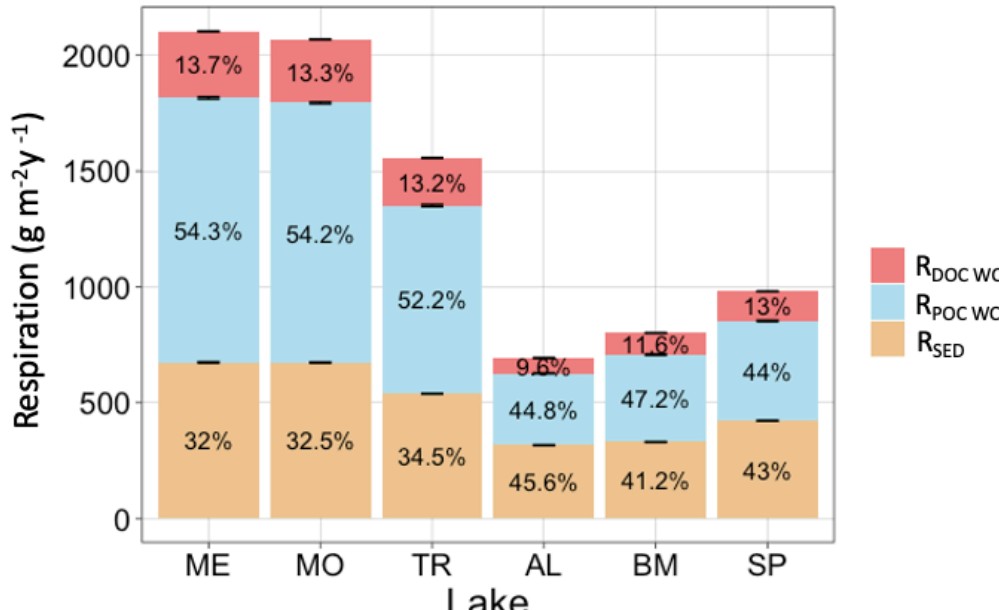

**Figure 8.** Total average annual hypolimnetic respiration, separated by percentages attributed to water column DOC ($R_{DOC\ WC}$), water column POC ($R_{POC\ WC}$), and sediment ($R_{SED}$) organic carbon sources. Standard error bars for the annual respiration values are indicated as well.

## 4 Discussion

### 4.1 Autochthonous and Allochthonous Loads

Autochthony was the dominant source of OC subsidizing hypolimnetic respiration in the

study lakes. The importance of autochthonous OC pools in ecosystem respiration was

surprising, given ample research highlighting the dominance of allochthonous OC in north

temperate lakes (Wilkinson et al. 2013; Hanson et al. 2011; Hanson et al. 2014). This

outcome emphasizes the utility of process-based models in studying mechanisms that discern



the relative contributions of different pools of organic matter to lake metabolism.
Autochthonous OC pools have higher turnover rates than allochthonous OC pools (Dordoni
et al., 2022) and often are lower in concentration than the more recalcitrant allochthonous
pools (Wilkinson et al. 2013). Thus, studies based on correlative relationships between lake
concentrations of organic matter and water quality metrics, likely overlook the importance of
more labile organic matter in driving observable ecosystem phenomena, such as gas flux and
formation of hypolimnetic anoxia (Evans et al., 2005; Feng et al., 2022). By quantifying
metabolism fluxes relevant to both OC pools, we can recreate shorter-term OC processes that
quantify high turnover of labile organic matter, which would typically be missed by
empirical studies based on monthly or annual observations.

Allochthony and autochthony are important to lake carbon cycling, but in ways that play out
at different time scales. Allochthonous OC has been well-established as an important factor
in driving negative NEP through a number of mechanisms (Wilkinson et al., 2013; Hanson et
al., 2014; Hanson et al., 2011). Allochthony contributes to water quality variables, such as
Secchi depth (Solomon et al. 2015), by providing the bulk of DOC in most lakes (Wilkinson
et al., 2013) and can drive persistent hypolimnetic anoxia in dystrophic lakes (Knoll et al.,
2018). In contrast, autochthony contributes to seasonal dynamics of water quality through
rapid changes in OC that can appear and disappear within a season. Within that seasonal time
frame, autochthonous POC settling from the epilimnion can drive hypolimnetic respiration,
thus controlling another key water quality metric, oxygen depletion. It is worth noting that
our model does not discern allochthonous and autochthonous sediment OC, however we





show that autochthonous OC makes up the largest proportion of OC loads in our study lakes
and therefore autochthony likely contributes substantially to the sediment OC pool. For
highly eutrophic lakes, the model results show excess autochthony stored in the sediments
which may carry into subsequent years, potentially providing additional substrate for
sediment respiration. Thus, understanding and predicting controls over hypolimnetic oxygen
depletion benefits from quantifying both allochthonous and autochthonous OC cycles.

Differences in trophic status, hydrologic residence time, and inflow sources help explain the
relative proportion of allochthonous versus autochthonous OC among lakes in our study.
Water residence times (Hotchkiss et al. 2018; McCullough et al. 2018) and surrounding land
cover (Hanson et al. 2014) have been shown to have a substantial impact on OC dynamics by
controlling allochthonous OC loading and NEP trends on lakes included in our study
(Hanson et al. 2014, McCullough et al. 2018). We built upon these ideas by recreating daily
watershed loading dynamics of POC and DOC from derived discharge data and incorporating
nutrient control over lake primary production by using high quality and long-term
observational data. The northern lakes are embedded in a forest and wetland landscape,
which are characteristic of having higher DOC than the urban and agricultural landscape of
the southern lakes (Creed et al., 2003). This creates variation in allochthonous loading across
the study lakes. Lake trophic state and productivity are a major control for autochthonous
production, which influences autochthonous loads across the study lakes as well. For lake
metrics that are comparable between studies, such as allochthonous loading and export,





allochthonous water column respiration, and total OC burial, our results were within 20% of
values in related studies (Hanson et al. 2014, McCullough et al. 2018).

**4.2 Hypolimnetic Respiration**
Given the importance of autochthonous POC to hypolimnetic respiration, we assume it
contributes substantially to both sediment respiration and respiration in the water column.
While previous work found that sediment respiration was the dominant respiration source for
lakes with depth ranges encompassed within our study (Steinsberger 2020), we found that
water column respiration was at least as important, if not more so. Differences in these
findings could be linked to uncertainty in the settling velocity of POC, due to lack of
empirical POC settling velocity measurements. Perhaps POC mineralized in the hypolimnia
of our modeled lakes passes more quickly to the sediments in real ecosystems, shifting the
balance of respiration more toward the sediments. It has been shown that POC respiration
contributes substantially to hypolimnetic DO depletion (Jenny et al. 2016), and POC settling
velocities can be highly variable, suggesting that assumptions around vertical distribution of
lake POC deserve further investigation. Another possible explanation for these differences
could be that our model missed allochthonous POC loads from extreme events (Carpenter et
al., 2012) which can increase the amount of legacy OC stored in the sediments and increase
sediment respiration. Our model also does not account for reduced respiration rates due to
OC aging, which may explain our higher values of water column respiration. Finally, our
model includes entrainment as a possible oxygen source to the hypolimnion, which must be
offset by respiration to fit observed hypolimnetic DO changes. Any study that underestimates





DO sources to the hypolimnion likely underestimates total respiration. Our findings highlight
the importance of autochthonous POC in hypolimnetic oxygen depletion and suggest that
related processes, such as the timing of nutrient loading, changes in thermocline depth, or
zooplankton grazing, could impact overall lake respiration dynamics and anoxia formation
(Schindler et al., 2016; Ladwig et al., 2021; Müller et al., 2012).

**4.3 Long-term Dynamics**
Although autochthonous OC dominated the loads across the study lakes, analysis of the long-
term OC dynamics supports the importance of allochthony in lakes. Net Ecosystem
Production (NEP) has been used to quantify heterotrophy and autotrophy in lakes (Odum
1956, Hanson et al. 2003, Cole et al. 2000, Lovett et al. 2006), and using this metric over
multiple decades allowed us to analyze long-term impacts of allochthony.  TR, BM, and SP
fluctuated between heterotrophy and autotrophy, usually in tandem with trends in hydrology,
which acts as a main control of allochthonous OC. This suggests that allochthonous OC
inputs may be less important for seasonal anoxia but can still drive a lake toward negative
NEP and contribute to sediment carbon storage over long time periods. ME, MO, and AL
tended to become less autotrophic over time (Fig. 6), a pattern that coincided with significant
decreasing trends in mean epilimnetic total phosphorus concentrations for two of the lakes,
ME and AL (SI Fig. 5). In our model NPP and phosphorus are directly related, so decreases
in phosphorus are likely to cause decreases in NEP. Short-term respiration of autochthonous
POC can account for rapid decreases in hypolimnetic DO, but allochthonous POC, which
tends to be more recalcitrant, provides long-term subsidy of ecosystem respiration that can



result in long-term net heterotrophy. Thus, it's critical to understand and quantify both the
rapid internal cycling based on autochthony and the long and slow turnover of allochthony.

Through a perspective that includes cycling of both allochthony and autochthony, we can
expand our conceptual model of metabolism to better understand time dynamics of lake
water quality at the ecosystem scale. Autochthony has pronounced seasonal dynamics,
typically associated with the temporal variability of phytoplankton communities and the
growth and senescence of macrophytes (Rautio et al., 2011). While allochthony can also have
strong seasonal patterns associated with leaf litter input, pollen blooms, and spring runoff
events, its more recalcitrant nature leads to a less pronounced seasonal signal at the
ecosystem scale (Wilkinson et al., 2013, Tranvik 1998). When considered together, it seems
that allochthony underlies long and slow changes in metabolism patterns, while autochthony
overlays strong seasonality. Both OC pools are important for ecosystem scale metabolism
processes, and their consequences are evident at different time scales. Therefore, the
interactions of both OC sources and their influences on water quality patterns deserve further
investigation.

Autochthonous OC control over hypolimnetic respiration should be a primary consideration
for understanding the influence of OC on ecosystem dynamics. Hypolimnetic oxygen
depletion and anoxia in productive lakes can be mitigated by reducing autochthonous
production of OC, which we show is mainly driven by nutrient availability. This study also
identifies the need for a better understanding of internal and external OC loads in lakes.



Previous studies have found heterotrophic behavior in less productive lakes, but our findings
highlight the importance of autochthony in these lakes, especially for shorter-time scale
processes that can be missed by looking at broad annual patterns. By using a one-
dimensional model, we are able to also understand how surface metabolism processes can
impact bottom layer dynamics, which would not be possible with a zero-dimensional model.
Looking forward, we believe that our understanding of these processes could be improved by
building a coupled watershed - metabolism model to more closely explore causal relations
between watershed hydrology, nutrient dynamics, and lake morphometry.





*Code Availability*
Model code and figure creation code are archived in the Environmental Data Initiative
repository (https://doi.org/10.6073/PASTA/1B5B947999AA2F9E0E95C91782B36EE9,
Delany, 2022).
*Data Availability*
Driving data, model configuration files, and model result data are archived in the
Environmental Data Initiative repository
(https://doi.org/10.6073/PASTA/1B5B947999AA2F9E0E95C91782B36EE9, Delany, 2022).
*Author Contributions*
AD, PH, RL, and CB assisted with model development and analysis of results. AD and PH
prepared the manuscript with contributions from RL, CB, and EA.
*Competing Interests*
The authors declare that they have no conflict of interest.

*Acknowledgements:*
Funding was provided through the National Science Foundation (NSF), with grants DEB-
1753639, DEB-1753657, and DEB-2025982. Funding for Ellen Albright was provided by the
NSF Graduate Research Fellowship Program (GRFP), and the Iowa Department of Natural
Resources (contract #22CRDLWBMBALM-0002). Funding for Robert Ladwig was
provided by the NSF ABI development grant (#DBI 1759865), UW-Madison Data Science
Initiative grant, and the NSF HDR grant (#1934633). Data were provided by the North
Temperate Lakes Long Term Ecological Research Program and was accessed through the
Environmental Data Initiative (DOI: 10.6073/pasta/0dbbfdbcdee623477c000106c444f3fd).



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
