# Peer review of "The Influence of Carbon Cycling on Oxygen Depletion in North-Temperate Lakes"

_EGUsphere, 2023_

## Author Comment (AC1)

[revised manuscript text omitted]

---

## Author Comment (AC2)

[revised manuscript text omitted]

---

## Author Response (AR1)

**Responses To All Reviewers**

We want to thank the reviewers for their helpful and detailed reviews of our manuscript. Each comment has been considered carefully, and we have made updates to our manuscript where applicable. All changes and updates have been outlined in this document. During the process of updating the manuscript, we discovered slight source code changes in our model and decided it was prudent to recalibrate the model parameters. Although overall patterns and main findings of the results remain unchanged, the new set of optimal model parameters resulted in a slight revision of visualizations throughout the manuscript.

**Major Changes To Methodology (Sensitivity, Calibration, and Uncertainty)**

Based on reviewer comments, we added a sensitivity analysis of the model parameters. We used the global sensitivity method from Morris (1991) to investigate the sensitivities of model output variables on each model parameter. The sensitivity analysis showed that there were nine parameters to which the model was consistently sensitive across the six study lakes. This group included the ratio of DOC and POC produced from NPP (C_npp), the maximum daily productivity parameter (Pmax), the inflow concentration of recalcitrant POC (POCR_inflow), the settling velocity of recalcitrant POC (K_POCR), the temperature fitting coefficients for productivity and respiration ($\theta$_npp, $\theta$_resp), the slope of the irradiance/productivity curve (IP), the sediment respiration flux (Rsed), and the respiration rate of DOCL (Resp_DOCL).

We chose a subset of the nine parameters to include in the uncertainty analysis based on the following justifications. The model results showed that recalcitrant substrates are of lesser importance for lake metabolism dynamics, so we chose not to further investigate the uncertainty of the POCR_inflow and K_POCR parameters. The Pmax and IP parameters are directly correlated, so we chose to remove Pmax from further uncertainty considerations. The $\theta$_npp and $\theta$_resp parameters act as substitutes for water temperature, a well-known "master variable" in water quality modeling, and directly reflect seasonality in the model. Therefore, we chose to omit these parameters for further uncertainty calculations. The final subset of parameters for uncertainty analysis consisted of C_npp, Resp_DOCL, R_sed, and IP. Of the four parameters, we felt C_npp was best constrained by the literature. To reduce the number of parameters estimated in the calibration process we restricted the automated constrained parameter search to the remaining three.

To clarify the text regarding model calibration we rewrote that section of the methods to better integrate the sensitivity analysis. There are three types of parameters – constants, manually calibrated based on literature, and constrained search based on the previous sensitivity analysis and literature ranges. Parameters with low sensitivity on model outcomes were manually

calibrated. To account for the high sensitivity of IP, Rsed, and Resp_DOCL, we ran an automatic constrained search across the joint parameter space to identify values that led to the best model fit. We needed to recalibrate the model parameters because of slight source code changes. Although overall result patterns and main findings remained unchanged, the new set of optimal model parameters resulted in a slight revision of visualizations throughout the manuscript.

Sensitivity guided uncertainty analysis. To quantify uncertainty around model predictions, we sampled IP, Rsed, and Resp_DOCL simultaneously from uniform distributions defined by literature values (Table 3). We plotted the 2.5 and 97.5% quantiles for these distributions and included them in the time series plots (Fig 2, 3, 4, SI Fig 1,2,3).

**Reviewer 1**

**Specific Comments:**

Line 79: It would be helpful to provide more detail on "external forcings" here. Which variables are you most considering? Hydrology, climate, nutrient inputs?

**Reply by authors:** We have updated the sentence to explain this in more detail:
      "An increase in the prevalence of hypolimnetic anoxia and associated water quality degradation in temperate lakes indicates the need to better understand how lake ecological processes interact with external forcings, such as hydrology and nutrient inputs, to lead towards the development of anoxia (Jenny et al. 2016 a,b)"

Line 88-96: Allochthonous OC can also impact DO dynamics through increasing thermal stratification due to its high chromophoricity. Is this process included in the model? Can you disentangle allochthonous OC influences from the effect of increased stratification versus directly fueling microbial respiration?

**Reply by authors:** The reviewer raises an interesting point. Stratification dynamics in the model are estimated from previously modeled temperature profiles for the study lakes. In the previously modeled temperature profiles, water characteristics controlling thermal stratification, such as chromophoric organic matter, would have been subsumed in other parameters during model fitting. We have no observational data for chromophoricity of allochthonous loads, and so the time dynamics of chromophoricity of allochthonous OC is not considered in our model.

However, the light extinction of DOC (LEC_DOC) is a parameter that we use in the model to calculate the total light extinction coefficient (K_LEC), which in turn is used to control NPP. We assume that the majority of DOC in the standing stock of the water column in a lake is from allochthonous sources. We assume that lakes with higher allochthonous loads likely have a higher chromophoric component to that load and we increase the LEC_DOC parameter to represent the effect that darker (newer) DOC has on water column light and the NPP of a lake (SI Table 5).

Figure 1: To improve clarity and simplicity of the model conceptual figure, consider removing the dashed boxes of TP and IC, as they are not state variables tracked in the model. Specifically, since IC is considered non-limiting and not a driver in any equation, the visualization would benefit from its removal. TP could be included by representing primary productivity as a function dependent on TP, ie. primary productivity(TP).

**Reply by authors:** We agree and have made these changes.

Table 3, eq. 20: Do recalcitrant and labile OC have different chromophoricity? They are treated similarly in their effect on light extinction coefficient in the model. Did you consider weighting recalcitrant OC more due to high aromaticity or light attenuating compounds? Would that change model dynamics?

**Reply by authors:** We do not account for chromophoricity differences between recalcitrant and labile OC. Given its labile nature, autochthonous DOC rarely accounts for more than 10-20% of the total DOC pool. DOC observed in a lake is mostly from allochthonous sources that are more recalcitrant in nature. While we can imagine very interesting short-term dynamics in water clarity as a function of, e.g., storm events, our focus in this manuscript was on longer term dynamics.

What about all the other parameters that were manually fit in Table 4. How were those fit and was any sensitivity analysis conducted?

**Reply by authors:** Please see our top reply regarding Major Changes to Methodology

Line 394: Free parameters were fit manually across their ranges, and it seems parameter values chosen were often at the extremes of the ranges. They also are strongly segregated across the northern and southern lakes, which may be driving many of the differences. How much did parameter value influence model fit? A sensitivity analysis here is needed, or atleast reporting the model residuals with different parameter values. Additionally, did you consider extending the ranges since chosen values were often at the extremes?

**Reply by authors:** Please see our top reply regarding Major Changes to Methodology

Lines 473-476: The authors report differences in the contributions of OC fates in the overall budget across the study lakes. Are these statistically significant differences? It would help interpretation to determine this, as the error bars on Figure 5 seem to be overlapping across lakes, but it is hard to compare across lake panels.

**Reply by authors:** We compared the contributions of OC sources and fates across the study lakes using the Wilcoxon statistical test for significance. Please see the attached table (SI Table 6) for these results. Note, any p-value less than 0.05 was considered significant. All non-significant differences between lakes are indicated by "NS". We found similarities in the flux values between ME and MO as well as BM and SP. This helps highlight that OC fluxes for lakes within the same trophic status were more similar and that OC fluxes for lakes across the trophic gradient were less similar.

Lines 580-583: What about anaerobic carbon metabolism? How would anaerobic metabolism influence sediment OC pools? The surplus of auto OC could be a direct reflection of sediment respiration nearing zero when oxygen is depleted. Previous research shows relationships between anaerobic OC degradation and autochthony, such that the OC could be mineralized as CH4, and there is a growing body of work on the importance of anaerobic metabolism for overall carbon cycling and OC burial in stratified lakes. I think it is worth noting that this is not considered and considering how it might change model predictions.

**Reply by authors:** We omitted anaerobic metabolism in the current modeling work to focus intensively on the relationship between carbon cycling and oxygen depletion. Nonetheless, as the reviewer has pointed out, anaerobic mineralization of organic carbon is an important biogeochemical process. Under low DO concentrations, anaerobic mineralization can account for

up to > 92 % of organic carbon respiration mostly through methanogenesis (Maerki et al. 2009), resulting in a release of methane into the water column. In Lake Mendota, previous research found that most dissolved methane is oxidized in the water column (Hart 2017), thus contributing to the overall oxygen demand. In our model, such oxidation is subsumed in overall estimates of respiration. What our model misses is ebullition of methane. We added a clarification to the Methods section 2.3.4 (Internal Lake Metabolism Fluxes):

"Note that we did not include anaerobic carbon metabolism in our modeling approach and discuss potential shortcomings in the discussion section"

Further, we added this paragraph to the discussion:

"Anaerobic mineralization of organic carbon is an important biogeochemical process and can be a substantial carbon sink through methanogenesis (Maerki et al. 2009). Although methanogenesis is not incorporated into our model, methane dissolved in the water column of Lake Mendota is mostly oxidized (Hart 2017), thus contributing to the overall oxygen demand, which is accounted for in our model. What remains unaccounted is ebullition of methane, which is a carbon flux that is difficult to quantify (McClure et al. 2020). Future metabolism studies that include these processes might find a decrease in annual OC burial rates. Although we believe that ebullition is not a substantial portion of the lake's carbon mass budget, that remains to be studied more carefully. As the model accounts for DO consumption through calibration, the overall flux would not change even if we link DO consumption to methane oxidation, only the process description would be more realistic."

**Technical Corrections:**

Line 176-192: Where was hydrology data from Lake MO obtained from? This lake is missing from the description.

**Reply by authors:** A description for Lake MO hydrology can be found in the manuscript (L264-266):

"ME is the predominant hydrologic source for MO (Lathrop & Carpenter 2014), thus, MO inflow is assumed to be equal to ME outflow, and MO outflow is assumed to be equal to MO inflow."

Table 3, Eq. 1: Missing open parentheses in front of NPP.

**Reply by authors:** Thank you for catching this. We have fixed this and it will be updated for the new version of the manuscript.

Additional References:

Maerki, Martin, Müller, Beat, Dinkel, Christian, Wehrli, Bernhard, (2009), Mineralization pathways in lake sediments with different oxygen and organic carbon supply, *Limnology and Oceanography*, 54, doi: 10.4319/lo.2009.54.2.0428.

Morris, M. D.: Factorial Sampling Plans for Preliminary Computational Experiments, Technometrics, 33, 161–174, https://doi.org/10.1080/00401706.1991.10484804, 1991.

**Reviewer 2**

**General Comments:**

The introduction struggles to highlight the need to connect the C and O cycles in lakes. Many sentences are vague and poorly informative, leaving the reader to check by themselves what is really said in the cited references. For instance, at different places, trophic status, hydrology, and morphometry are cited as all acting/influencing deep hypoxia (l74,l92,l100, l107, l109). But how they are mechanistically acting upon hypoxia remains very vague, and readers should go and check the references. Instead of mentioning that factors 'influence'' hypoxia, I would find it much more informative to explain how they do, providing more informative facts. I would suggest rewriting and refocusing the introduction, enriching the conceptual background. The main achievement of this paper is to show that heterotrophy does not necessarily generate greater hypoxia. There is a body of research relating terrestrial organic matter inputs to lake respiration; then, an easy set of hypotheses would be to drive from this. With a greater allochthony and heterotrophy, greater respiration could shift the oxygen balance towards depletion. Then, I suggest focusing on the processes acting upon hypolimnetic oxygen, namely, those by which oxygen is consumed and then relates to carbon sources, and those by which oxygen is renewed. It would be finally easier, from processes, to introduce how lake hydrology and morphometry can affect both consumption and renewal and finally modulate what to expect regarding oxygen depletion.

**Reply by authors:** The reviewer makes excellent suggestions for reformulating the Introduction to the manuscript. In keeping with the suggestions, we shift the emphasis to the possible role of autochthony to hypolimnetic respiration as an advancement to the oft-cited paradigm of net heterotrophy being the norm for lakes, with the implication that gradients of allochthony should explain gradients of hypolimnetic oxygen depletion. We provide the following topic sentences for a new Introduction, and we hope to have the opportunity to share the entire Introduction should we move to the next stage of the publication process.

1. Hypolimnetic oxygen depletion is a persistent and global phenomenon that degrades lake ecosystems services.
2. Allochthonous organic carbon (OC) loading to lakes explains the prevalence of negative net ecosystem production (i.e., net heterotrophy) and provides substrate for hypolimnetic oxygen depletion.
3. The contributions of OC from autochthony to hypolimnetic oxygen depletion may be important as well, despite its low concentrations relative to that of allochthonous OC in many lakes.

4. Understanding the relative importance of autochthony and allochthony to hypolimnetic oxygen depletion requires consideration of a number of physical and biological processes controlling oxygen sources and sinks in lakes.
5. The availability of long-term observational data combined with process-based models provides an opportunity to investigate OC sources and their control over the dynamics of lake DO across multiple time scales.
6. In this study, our goal is to investigate OC source contributions to lake carbon cycling and hypolimnetic oxygen depletion. We address the following questions:
    a. What are the dominant sources of organic carbon that contribute to hypolimnetic oxygen depletion, and how do their contributions differ across a group of diverse lakes over two decades?
    b. How do lake trophic state, hydrology, and morphometry influence the processing and fates of organic carbon loads in ways that affect hypolimnetic dissolved oxygen?

I still have not understood how allochthonous loadings were computed. Discharges come essentially from a hydrological model, but what about concentrations?

**Reply by authors:** Our process for computing allochthonous loads for the study lakes is described in L255-259 and L267-270. However, we believe that the reader would benefit from a more detailed description and have updated Methods section 2.3.2 (External Lake and Environmental Physical Fluxes) with the following information:

"ME allochthonous loads are based on observed allochthonous DOC and POC concentration ranges found in Hart et al. (2017) and were verified against the modeled OC equilibrium for the lake. MO inflow concentrations for DOC and POC were taken from the surface water concentrations of ME and were verified against load estimates in McCullough et al., 2018. Northern lake total OC inflow concentrations were taken from estimated annual loads found in Hanson et al., 2014. Specifically, DOC inflow concentrations were calibrated from winter in-lake observational data. POC inflow concentrations were calibrated from the remaining OC inflow budget for lakes."

There is also sometimes confusion between what is a flux and what is a rate. For instance, l303, "sediment respiration for the hypolimnion [...] is a constant daily rate", rate should be included after respiration.but l 332, "the respiration rates are also scaled by oxygen availability", here rate should be replaced by "flux"

**Response by authors:** This was an oversight on our part, and thank you for bringing it to our attention. We have fixed this in the manuscript.

l390, acronyms for r_sed and R_DOCL are different (Resp_DOCL and Resp_sed)

**Response by authors:** We have changed these acronyms to make them consistent throughout the manuscript.

**Specific Comments:**

R: l69-74. It is surprising to start the introduction by mentioning that hypoxia can lead to increased OC release from the sediment. At the same time, this process is not included in the very model aimed at linking the O and C cycle. The study of drivers of hypoxia in lakes could be better motivated.

**Response by authors:** We have removed this sentence from the introduction.

Table 3: r_rate not defined

**Response by authors:** We have added the following text to the Table 2 description:

"The term ($r_{rate}$) is included in Eq. 13 to represent the respiration rates of the different OC pools. It is included to simplify the table of equationsTerms not defined here are included in Table 3."

Results: RMSE is the only metric used to assess the goodness of fit. However, it poorly captures the model's ability to reproduce the seasonal dynamics, while this is the critical requirement for the model (as it is reinitialized each Jan 1st). Does it make sense to add another metric?

**Response by authors:** In addition to RMSE, we have calculated the Nash-Sutcliffe model efficiency coefficient (NSE) and the Kling-Gupta Efficiency (KGE) for the model state variables. These values are presented in SI Table 7.

L615- "It has been shown that POC respiration contributes substantially to hypolimnetic DO depletion (Jenny et al. 2016)," This quote needs to be checked. I do not recall Jenny et al quantifying the respiration of POC.

**Response by authors:** The reviewer has identified an oversight in our referencing the literature. We have updated the quote:
"POC respiration can contribute substantially to hypolimnetic DO depletion in both lakes and reservoirs (Beutel, 2003)"

L658-662. Is the long-term effect of allochthony testable from a model that uses annual reinitialization?

**Response by authors:** The model states are initialized at the first time step of the model run, and are not reinitialized at any point. We do include annual stratification and mixing dynamics, which are estimated using temperature profiles to calculate water column buoyancy. We have updated section 2.3 of the manuscript ("The Model") to clarify this point. See below:
"Throughout each year, the model tracks state variables and fluxes in the lake for each day (Fig. 1). These state variables are initialized at the first time step of the model and include DO and the labile and recalcitrant components of particulate organic carbon (POC) and dissolved organic carbon (DOC).

Additional References:

Morris, M. D.: Factorial Sampling Plans for Preliminary Computational Experiments, Technometrics, 33, 161–174, https://doi.org/10.1080/00401706.1991.10484804, 1991.

Beutel, Marc (2003) Hypolimnetic Anoxia and Sediment Oxygen Demand in California Drinking Water Reservoirs, Lake and Reservoir Management, 19:3, 208-221, DOI: 10.1080/07438140309354086

**General Comments:**

I could not determine whether the metabolism model was specifically developed for this analysis or whether it was based on some previously published model, as no citations were offered. If the model is specific to this analysis, citations should be provided that justify different modeling choices.

**Reply by authors:** The model was coded specifically for this analysis; however, many of the assumptions around model complexity and mathematical formulations were borrowed from literature cited. We chose to develop our own process-based model rather than use an existing model (such as GLM, Simstrat, etc.) so that we could simulate and measure the specific metabolism fluxes related to our study questions. As the reviewer points out, there are some important differences. We clarify these points through additional annotations and citations in the manuscript, as well as an additional paragraph written to clarify the issue.

I couldn't find a citation for the model linking TP to NPP (Equation 12). Where did this come from?

**Reply by authors:** Total phosphorus concentration in a layer is from observational data for each lake interpolated to the daily time scale. The interpolated values are then normalized for each individual lake to drive NPP. These values are standard-normal transformed for the entire time series; thus, the mean and variance of phosphorus is subsumed in the estimates of the IP and Pmax parameters. The time dynamics of normalized phosphorus concentrations are retained to represent seasonal P dynamics in the lake.

What is Pmax?

**Reply by authors:** The Pmax parameter is the maximum daily productivity for a lake and is based on trophic specific ranges provided in Wetzel (2001). These values are provided in units of [mgC/m2/day], and subsume lake-specific nutrient concentrations. We convert these values to units of [gC/m3/day] for use in our model.

How was this DOC allocated to labile and recalcitrant fractions? I'm guessing that it was assigned entirely to the recalcitrant fraction, but no information is provided.

**Reply by authors:** Allochthonus DOC loads were assigned entirely as recalcitrant substrates. To clarify this point for the reader, we have included the following description in Methods section 2.3.2 (External Lake and Environmental Physical Fluxes):

"For the northern lakes (TR, AL, BM, SP), we assume that allochthonous OC loads consist of entirely recalcitrant substrates and calibrate these and our recalcitrant OC export values from Hanson et al. (2014)."

Also, the presence of a INDOCL term in Equation (2) suggests that at least some allochthonous load is designated as labile.

**Reply by authors:** We assume that allochthonous DOC loading is a recalcitrant substrate, with the exception of Lake Monona (MO) where external OC loads are estimated from OC in the water column of Lake Mendota (ME), which is a mix of labile and recalcitrant OC. Given this relationship, we account for labile POC and DOC loading for MO and include additional terms for INDOCL and INPOCL for Equation 2 and 4 in Table 2, respectively.

Recalcitrant vs. labile OC. The focus of this paper is to determine the relative contributions of allochthonous and autochthonous OC to oxygen depletion, and the lability of OC is a crucial determinant of these contributions. As stated by the authors, autochthonous OC tends to be more labile than allochthonous OC, but as far as I can tell, all allochthonous OC is assigned to the recalcitrant fraction and all autochthonous OC is assigned to the labile fraction. If this is so, then isn't the conclusion that most oxygen depletion is driven by autochthonous OC self-evident? Why bother running the model?

**Reply by authors:** The total respiratory flux is the product of the respective labile and recalcitrant pool sizes and their decay rates, the available oxygen, and the ambient temperature. The oxygen demand, therefore, depends very much on the OC pool sizes, which vary by seasons and differ between thermal strata, and the ambient conditions, which vary seasonally and vertically. Based on ecosystem observations alone, the pool size of the recalcitrant fraction of OC is higher in many lakes, suggesting that ecosystem respiration may scale with allochthony. In other words, the importance of labile OC is easily missed because of its relatively small fraction within the larger OC pool. Without quantifying primary production and its fate, we have no way to account for its contribution to total respiration. How OC moves through the lake matters, as well. For example, it is conceivable that primary production in the epilimnion could have been offset by epilimnetic respiration, thus eliminating the autochthonous pool before it had a chance to contribute to hypolimnetic respiration. However, through modeling, we were able to quantify the autochthonous OC production and its export from the epilimnion to the hypolimnion and its total contribution to the organic carbon cycle, including the effects on oxygen consumption and the seasonality of those dynamics. These ideas are emphasized in the rewriting of the Introduction and in additional points made in the Discussion.

Model uncertainty. More work is needed to quantify the uncertainty in the model. The uncertainty in the key components of the model (estimating NPP and external OC loads) is likely large, but no information is offered as to how this uncertainty would affect model results. Assuming that the uncertainty of all model parameters is 20% of the mean value is also a gross simplification.

**Reply by authors:** Please see our top reply regarding Major Changes to Methodology.

**Line-Specific Comments:**

Table 1: The authors state that the sampled lakes cover a broad range of allochthonous loads, but the range of DOC concentrations is narrow relative to concentrations observed at the continental scale (see for example, values reported in EPA's National Lakes Assessment. It would be good to place the DOC concentrations in a broader context.

**Reply by authors:** The reviewer is correct. DOC concentrations of the study lakes represent a relatively narrow range of what has been observed in northern Wisconsin (Hanson et al. 2007), although they are typical for non dystrophic lakes in the region. The allochthonous loads, on the other hand, span about one order of magnitude across the lakes. The text has been modified as follows::

Although the full range of DOC concentrations for lakes in northern Wisconsin varies from about 2 to >30 mg L$^{-1}$ (Hanson et al. 2007), DOC concentrations among our study lakes covered a relatively narrow range typical of non dystrophic lakes in Wisconsin (Hanson et al. 2007) and are near the global averages previously estimated, i.e., 3.88 mg/L (Toming et al. 2020) and 5.71 mg/L (Sobek et al. 2007), respectively.

Line 190-192: The citation here points to documentation for the data but how exactly were nutrient concentrations used to computed PP?

**Reply by authors:** We compared our coefficients for primary production with those found in (Wetzel, 2001), which are in the units [mgC/m2/day]. Note that such coefficients are not per unit phosphorus, but rather, subsume lake-specific nutrient concentrations. To compare our value with theirs, we used the standard normal transform of observed P (i.e., removed the mean and divided by the standard deviation.) Those two statistical features of P become subsumed in the estimates of IP and Pmax, but the time dynamics of normalized P are retained to represent seasonal P dynamics in the lake.

Line 204: The Hipsey et al. 2019 paper is cited repeatedly but in my reading, this paper only covers the hydrodynamic part of the lake model and says nothing about metabolism models. Is the citation incorrect, or is there a different paper that should be cited?

**Reply by authors:** The reviewer has identified an oversight in our referencing the literature. We have updated the source to Hipsey et al. 2022.

Hipsey, M.R., (ed.). 2022. Modelling Aquatic Eco-Dynamics: Overview of the AED modular simulation platform. Zenodo. https://doi.org/10.5281/zenodo.6516222.

Line 257: What does the sentence here mean? It seems to state the same information as the previous sentence. (included below)
L257 – "We specifically use the allochthonous OC load values in this study to assist in the manual calibration of inflow recalcitrant POC and DOC concentrations for each lake"

**Reply by authors:** This sentence will be removed as it is redundant.

Line 266 - 267: How is the calibration performed? The citation again just documents the source of the data.

**Reply by authors:** This was an oversight on our part. Allochthonous OC concentrations were not calibrated. These values were taken from observed data in Hart et al. (2017), and we verified these values by back-calculating inflow concentrations based on the modeled OC equilibrium for the lake.

Line 319: So, all OC from NPP is assumed to be labile, right?

**Reply by authors:** Yes, this assumption is accurate. We have updated this sentence to reflect our assumptions.

"All OC derived from NPP is assumed to be labile and is split between particulate and dissolved OC production, with eighty percent produced as POC and twenty percent produced as DOC. This ratio was determined through model fitting and is similar to previously reported values (Hipsey et al. 2022).

Line 350: Is inorganic suspended sediment negligible in these lakes? Most Secchi computations would include a contribution from inorganic sediment.

**Reply by authors:** Due to the scarcity of long-term TSS data for these lakes, we have taken this into our model assumptions. We assume that TSS inorganic particulates are low in concentration and are a small percentage of the background water clarity. Given this, we do not include the impacts of inorganic TSS on Secchi in our model.

Line 406: This approach for Secchi seems very ad hoc. Why are Secchi's in northern lakes so variable? Higher sampling error? More temporal variability?

**Reply by authors:** We do not know why observed Secchi data are highly stochastic. We confirmed that characteristic in the observational data using time series decomposition. While there is an annual signal of about +/- 0.5 m, the noise component can be up to four times greater. As an example we have attached a time series decomposition for TR to this reply (SI Fig 7).

Line 442: The differences between model predictions of DOC and observed values are so great that it's hard to argue that the model is accurately reproducing temporal changes in DOC. Only predictions TR and SP are close to observations and that seems more due to the fact that the magnitude of DOC temporal variations is small.

**Reply by authors:** We recognize that there are small biases in the long term values of the prediction of the doc. It is also the case that the variance is not fully reproduced by the model. We document the goodness of fit more fully in SI Table 7 for the benefit of the reader.

Line 561: Good point. I'm struggling to determine whether the model described in this manuscript is accurate enough to support this statement. A more realistic sensitivity and uncertainty analysis would help, particularly if that uncertainty was carried through to examine the effects on the budgets depicted in Figure 5.

**Reply by authors:** We do not assert that our model outcomes prove this point but rather support it. We did add a sensitivity analysis which is described in the top reply. Our analysis of the OC budgets helps support this (SI Table 6).

Additional References:

Hipsey, M.R., (ed.). 2022. Modelling Aquatic Eco-Dynamics: Overview of the AED modular simulation platform. Zenodo. https://doi.org/10.5281/zenodo.6516222.

Morris, M. D.: Factorial Sampling Plans for Preliminary Computational Experiments, Technometrics, 33, 161–174, https://doi.org/10.1080/00401706.1991.10484804, 1991.

---

## Author Response (AR2)

We would like to thank the reviewers for their useful and detailed reviews of our manuscript. Please see below for our responses to each reviewer comment.

**Reviewer 2**

The authors made a good work while revising the paper.
I appreciated that they restructured the introduction, but I think some work is still required to limit overstatements.
I have a few specific comments below, but my main criticism, which is an easy fix, is that the authors justify their work by stating that "the relative importance of autochthonous and allochthonous OC in depleting oxygen across diverse lake systems remains uncertain" (Abstract). Similarly, they present their main finding, that hypolimnetic oxygen consumption comes mostly from the respiration of autochthonous OC, as surprising (l598-599).
Although I find that the study is providing a nice mechanistic understanding of the links between the C and O cyclings in lakes, the authors must tone down on the "lack of knowledge" in the intro and the "surprise" of the conclusion. The link between nutrient enrichment and oxygen depletion at lake bottoms is certainly not novel, so the fact that excess PP fuels oxygen depletion in hypolimnion is undoubtedly not a novelty. Already Odum in 1956 was on this line. I suggest the authors return to Nürnberg, Gertrud K., (1995), Quantifying anoxia in lakes, Limnology and Oceanography, 40, doi: 10.4319/lo.1995.40.6.1100. to justify their study while not undermining the basic limnological knowledge. Muller et al, 2012 also is a good reference on which they can build (already in the ref list).

Response: In the Abstract we replaced "relative importance" with "relative contribution" and rewrote the sentence. Similarly, in the 4th paragraph of the Introduction, we replaced "importance" with "contributions." We have also included the following sentence in the final paragraph of the introduction:

The importance of excess primary production to anoxia has been established (Nürnberg et al., 1995; Müller et al., 2012). We build upon this research by quantifying the timing and magnitude of OC contributions to hypolimnetic anoxia.

Specific comments

R1: L255-257
"We choose to develop our own model instead of GLM or Simstrat to … metabolic… " . Both Simstrat and GLM are physical models. Besides, just below, you mention that the GLM was used to simulate temperatures. So my guess is that you did not use available water quality models such as AED 2 for the C/O compartments

Response: We apologize for the confusion. We meant to say that we developed our own "water quality" model, and that we used the output of GLM for physical dynamics. The GLM output we used for temperature profiles was previously published (Read et al. 2021). We have clarified this in the manuscript.

R2: Table 3: duplicated line for the Respiration rate of POCR

Response: Thank you for pointing this out. We have fixed this error in the manuscript.

R3: Fig 6 and l 537: how much can you say from the lake metabolism given the uncertainties around the model, which are usually bigger than the difference between GPP and R? I think some care is required here.

Response: We have added the following text to section 3.2 of the manuscript:

"It is worth noting that our interpretation of metabolism dynamics in the results are based on the median NPP and Respiration flux values produced by the model. Because of the high uncertainty associated with these fluxes, we should be cautious about asserting inferences about long term changes in trophic state."

R4: L557 "Water column respiration contributes more than sediment respiration to total hypolimnetic respiration in the southern lakes compared to the northern lakes, with the exception of TR, where cumulative water column respiration is much larger than cumulative sediment respiration"
This sentence is quite circumvoluted. Maybe"Water column respiration contributes more than sediment respiration to total hypolimnetic respiration in the deepest lakes.

Response: We agree with this suggestion and have reflected this change in the manuscript.

R5: L 604 "Autochthonous OC pools have higher turnover rates than allochthonous OC pools (Dordoni et al., 2022) and often are lower in concentration than the more recalcitrant allochthonous pools (Wilkinson et al. 2013)." In the study lakes, autochthonous DOC dominates by far.

Response: We have added further text to the Discussion section (copied below), to help clarify our point. We also note that the lakes in our study (specifically, AL, BM, SP, and TR) were in the Wilkinson et al (2013) study.

New text added to section 4.1 of the manuscript:
"Similar to what was found by Wilkinson et al (2013), the standing stock of DOC in the water column of lakes in our study was from predominantly allochthonous sources. However, we emphasize in our study that autochthonous OC pools have higher turnover rates than allochthonous OC pools (Dordoni et al., 2022) and often are lower in concentration than the more recalcitrant allochthonous pools (Wilkinson et al. 2013)."

R6: I need help understanding Figure 8. Is the total average annual hypolimnetic respiration per lake different from the annual respiration flux presented in Figure 5? If it does not, why is there an order of magnitude of difference between fluxes (approx 40-150 g/m2/yr in Fig 5 and 400-1300 g/m2/yr in Fig 8)? If it is different, can you clarify?

Response: Fig 5 shows the annual whole-lake respiration flux (epi and hypo), and describes the flux over the entire lake surface area. Fig 8 shows the total annual lake respiration for just the hypolimnion, described over just the area of the hypolimnion (lake sediments). We estimated the area of the hypolimnion from the area of the thermocline. This difference of respiration pool (gC) and area (m2) results in different flux magnitudes (gC/m2).

**Reviewer 3**

The revised manuscript improves significantly on the earlier draft. However, there is still no information regarding the basis of the model linking NPP to TP. Furthermore, the author's response to my original comment does not mention a citation, nor does it really address my comment regarding the basis of this model formulation. The citations that are mentioned in the manuscript describe NPP models based on chl a (McCullough et al. 2018) and based on nitrate and phosphate concentrations (Ladwig et al. 2021, Hippsey et al. 2022). Both of these parameters provide a better basis for estimating NPP than TP, which is used in the current study. Total phosphorus includes contributions from sediment-bound P, so there is likely to be variability in TP that is not associated with changes NPP. I suspect that data limitations dictated the use of TP in the NPP model, and if so, the manuscript should make this clear. Furthermore, lacking a citation for a TP-NPP model, the authors should provide some justification for why this model formulation is valid.

Response: The reviewer highlights an important issue, which is the relevance of total phosphorus to primary production in the water column. We have added the following text into section 2.3.4 or the manuscript:

"For our most eutrophic lake, Lake Mendota, most of the TP in Lake Mendota's water column is from internal loading (Soranno et al.1997) as a consequence of mineralization of organic forms of phosphorus (principally from phytoplankton) or released in mineral form from sediments (Hoffman et al. 2013). As shown by Read et al. (2014), most of the total phosphorus in the water column is in reactive forms – either dissolved reactive or as part of phytoplankton biomass. Considering phosphorus can cycle rapidly among forms, and considering the high correlation between TP and reactive forms of phosphorus (Read et al., 2014), we consider total phosphorus to be a suitable index of available nutrients for primary production. This simplification obviates the need for additional processes in the model for cycling phosphorus among various forms, which reduces model data requirements and the need for more complex calibration. Lakes in our study have relatively low sediment loading, so we assume tightly bound phosphorus in the water column (i.e., phosphorus generally unavailable to biotic processes) is a small fraction of the total phosphorus. In lakes with high sediment loads, the assumption that TP is a reasonable index for NPP may not hold true."

R1: Line 208: How were these discharge discrepancies quantified?

Response: We quantified these discrepancies by comparing the modeled discharge estimates to published water residence time values for our study lakes. We have added the following text to help clarify this point:

"After comparing the modeled discharge data with published water residence times for our study lakes (Table 1), we found that the derived discharge data for ME, TR, AL, and SP was approximately 20-50% higher than previously reported values (Hunt et al. 2013, Webster et al. 1996), depending on the lake, while hydrology in BM was approximately 25% too low (Hunt et al. 2013)"

R2: Line 254: Should this be Ladwig (2021) or is there a 2022 publication not listed in the reference list?

Response: Thank you for pointing out this oversight on our part. We have fixed this issue in the manuscript

R3: Table 3: Respiration rate for POCR listed twice.

Response: Thank you for pointing this out. We have fixed this error in the manuscript.

R4: Table 3: Why do POCL and POCR have different settling rates? Is there a reason to expect this difference?

Response: For POCR, we note that the settling rate [1.2 m/d] equates approximately to a first order decay of 0.1 1/d, when divided by the mean depth of the lake (e.g., 1.2 m/d / 12.8m ~= 0.1 m/d). This rate is roughly 2 orders of magnitude greater than the decay rate for POCR. Thus, the model is not particularly sensitive to the POCR settling rate. The POCR represents leaves, pollen, detritus, etc. from the catchment. POCL represents the community of phytoplankton in our model, which may include fast-sinking diatoms and blue-greens that sometimes have buoyancy. We cite a diversity of sinking rates in the manuscript. It seems reasonable that POCL would sink more slowly than POCR.

R5: Line 346: should this read: "…in a layer is taken from…"
Response: Yes, we agree and have reflected these changes in the manuscript.

R6: Line 418: Move descriptions of "constant", "manually calibrated", and "parameters calibrated…" earlier in the manuscript so that the reader understands these categories when looking at Table 3.

Response: Thank you for this suggestion. We agree that Table 3 should be closer to the model parameter group descriptions. Rather than move the descriptions further up in the manuscript, we have moved Table 3 down further closer to the Model Sensitivity and Parameter Calibration section.

R7: Line 424: Which observations were used for the model fit? DO? DOC? POC? Secchi? Were all groups of observations equally weighted so that the parameters were estimated to maximize model fit to all measurements? If so, wouldn't it make more sense to more heavily weight predictions of DOC and POC, given the focus of this analysis?

Response: While the reviewer makes an interesting point about weighting DOC and POC in model fits, we note that DOC data are somewhat sparse and not especially dynamic, and that

Secchi is our index for POC. We have high confidence in values of DO, and DO is critical to calibrating metabolic rates. Thus, we do not feel a sufficient justification for weighting more heavily any one observational variable.

The following text was included in the original manuscript submission but was removed:

During the model fitting, errors in modeled DO, DOC, and Secchi depth are weighted equally in the southern lakes. In the northern lakes, fitting Secchi depth was challenging. Initial model fits revealed that patterns in observed Secchi did not show regular seasonality and were highly stochastic. Therefore, we use a moving average on observational data and predictions of Secchi depth and calculate the residuals as the difference between the two averaged time series. This is done to remove stochasticity from the observational data and fit the model predictions to the average observed Secchi value. We use a moving average window of 15 observations because we want to capture the average annual Secchi depth trend, and there are roughly 15 observations per year.

We have updated section 2.4 to include some of this original text (copied below):

During the model fitting, errors in modeled DO, DOC, and Secchi depth are weighted equally in the southern lakes. Secchi depths in the northern lakes were highly stochastic, and therefore we use a moving average on observational data and predictions of Secchi depth and calculate the residuals as the difference between the two averaged time series. We use a moving average window of 15 observations because we want to capture the average annual Secchi depth trend, and there are roughly 15 observations per year.

R8: Figure 2: Hypolimnetic DO depletion rate seems notably off in SP. Any reason why?

Response: The hypolimnetic DO depletion rate for SP is higher than observed in some years and lower in others, indicating more uncertainty. We can only speculate about the cause. The highly uncertain DO depletion rate may have to do with the morphometry of SP, which is more of a flat-bowl-shape than the rest of the northern study lakes and therefore may have a higher proportional sediment area. Although lake hypsometry, along with thermal profile, controls the volume of hypolimnion in contact with sediments in our model, there may be other factors related to morphometry (e.g., sediment focusing) that remain unaccounted.

We have added additional text to section 4.2 to help clarify this point.

R9: Figure 3: It seems like the amplitude of variation in DOC (difference between the peak of each seasonal cycle and the trough) is overestimated in all lakes except for BM. I would guess that the majority of this seasonal variation in DOC is due to autochthonous sources, so could this difference in amplitude indicate that the predicted autochthonous contributions to DOC are too high? If so, this would affect the relative importance of OC sources presented in later figures (e.g., Line 512). I think the conclusions regarding the importance of OC sources would be stronger if the amplitudes of variations of observed vs. predicted were more similar (see comment for Line 424 regarding model fitting).

Response: As the reviewer points out, most of the DOC dynamics are from autochthony. In our model, autochthony is split between dissolved and particulate fractions. Lowering the annual amplitude of modeled DOC would perhaps raise the annual amplitude of modeled POC, but the OC would still be autochthonous. Most autochthony is respired in the water column in our lakes, regardless of its form (Fig. 5). We also note that BM, pointed out by the reviewer as the lake with the lowest DOC annual amplitude, does not have OC fates that are remarkably different from other similar lakes in our study (Fig. 5). Although recalibrating the model always changes numbers, we believe the differences would be minor and would not alter our conclusions.

R10: Line 581: Isn't the smaller proportion of total respiration attributed to DOC a direct consequence of the fact that selected values for r for particulates is set to 5x that of dissolved values? So, doesn't this finding simply reflect model assumptions?

Response: Relative rates do, of course, have an effect on fluxes and standing stocks. However, there is no way to accurately predict a priori what will happen at the ecosystem scale. The total respiratory flux is the product of the respective labile and recalcitrant pool sizes and their decay rates, the available oxygen, and the ambient temperature. The oxygen demand, therefore, depends very much on the OC pool sizes, which vary by seasons and differ between thermal strata. Based on ecosystem observations alone, the pool size of the recalcitrant fraction of OC is higher in many lakes, suggesting that ecosystem respiration scales with allochthony. Without quantifying primary production and its fate, we have no way to account for its contribution to total respiration. How OC moves through the lake matters, as well. For example, it is conceivable that primary production in the epilimnion could have been offset by epilimnetic respiration, thus eliminating the autochthonous pool before it had a chance to contribute to hypolimnetic respiration. However, through modeling, we were able to quantify the autochthonous OC production and its export from the epilimnion to the hypolimnion and its total contribution to the organic carbon cycle, including the effects on oxygen consumption and the seasonality of those dynamics.

R11: Line 597: I think this approach has merit, and this statement may well be true, but I also think it makes sense to place this statement in the context of the different model assumptions.

Response: The criticism raised here is of a similar nature to previous criticisms about being careful not to inflate the importance of the findings. We agree that the text could be "toned down" and tightened and edited the first Discussion paragraph accordingly.

R12: Line 609: Are there DOC datasets available that are collected more frequently? It would be interesting to validate against more intense data.

Response: We wish there were more DOC observational data available for this study, and we agree that would make for an interesting comparison with our results. Although it is outside of the scope of this manuscript, we agree that a field campaign focusing on alternative DOC pools and their sources in lakes would be a great topic for future research.